# Upregulated expression of ubiquitin ligase TRIM21 promotes PKM2 nuclear translocation and astrocyte activation in experimental autoimmune encephalomyelitis

**Luting Yang[†], Chunqing Hu[†], Xiaowen Chen[†], Jie Zhang[†], Zhe Feng, Yanxin Xiao, Weitai He, Tingting Cui, Xin Zhang, Yang Yang, Yaling Zhang, Yaping Yan\***

Key Laboratory of the Ministry of Education for Medicinal Resources and Natural Pharmaceutical Chemistry, National Engineering Laboratory for Resource Development of Endangered Crude Drugs in Northwest of China, College of Life Sciences, Shaanxi Normal University, Xi'an, China

**\*For correspondence:**
yaping.yan@snnu.edu.cn

[†]These authors contributed equally to this work

**Competing interest:** The authors declare that no competing interests exist.

**Abstract** Reactive astrocytes play critical roles in the occurrence of various neurological diseases such as multiple sclerosis. Activation of astrocytes is often accompanied by a glycolysis-dominant metabolic switch. However, the role and molecular mechanism of metabolic reprogramming in activation of astrocytes have not been clarified. Here, we found that PKM2, a rate-limiting enzyme of glycolysis, displayed nuclear translocation in astrocytes of EAE (experimental autoimmune encephalomyelitis) mice, an animal model of multiple sclerosis. Prevention of PKM2 nuclear import by DASA-58 significantly reduced the activation of mice primary astrocytes, which was observed by decreased proliferation, glycolysis and secretion of inflammatory cytokines. Most importantly, we identified the ubiquitination-mediated regulation of PKM2 nuclear import by ubiquitin ligase TRIM21. TRIM21 interacted with PKM2, promoted its nuclear translocation and stimulated its nuclear activity to phosphorylate STAT3, NF-κB and interact with c-myc. Further single-cell RNA sequencing and immunofluorescence staining demonstrated that TRIM21 expression was upregulated in astrocytes of EAE. TRIM21 overexpressing in mice primary astrocytes enhanced PKM2-dependent glycolysis and proliferation, which could be reversed by DASA-58. Moreover, intracerebroventricular injection of a lentiviral vector to knockdown TRIM21 in astrocytes or intraperitoneal injection of TEPP-46, which inhibit the nuclear translocation of PKM2, effectively decreased disease severity, CNS inflammation and demyelination in EAE. Collectively, our study provides novel insights into the pathological function of nuclear glycolytic enzyme PKM2 and ubiquitination-mediated regulatory mechanism that are involved in astrocyte activation. Targeting this axis may be a potential therapeutic strategy for the treatment of astrocyte-involved neurological disease.

## eLife assessment

This **important** work describes the activation of astrocytes via the nuclear translocation of PKM2 in an animal model of multiple sclerosis. This study provides **convincing** evidence of the interaction between TRIM21 and PKM2 as the crucial molecular event leading to the translocation of PKM2 and a metabolic shift towards glycolysis dominance, fostering proliferation in stimulated astrocytes. This finding is significant as it underscores the potential of targeting glycolytic metabolism to mitigate

neurological diseases mediated by astrocytes, offering a strong rationale for potential therapeutic interventions.

## Introduction

Multiple sclerosis (MS) is a chronic inflammatory disease of the CNS, which accounts for the leading cause of neurological disability in young adults. The hallmarks of this disease is varied and complex, ranging from astrocyte proliferation, microglia activation, neuroinflammation and damage to myelin sheaths (*Kuhlmann et al., 2023*). Accumulating evidence suggests the critical roles of neurons in MS pathology. However, with the deepening of research, local glia cells have been shown to potentiate inflammation and lead to neurodegeneration, among which astrocytes have attracted much attention with their diverse functions (*Lee et al., 2023*).

Astrocytes are the most abundant type of glia cells and provide physical, structural, and metabolic support for neurons. Astrocytes respond to CNS diseases through a process of activation that encompasses cell proliferation, morphological, molecular and functional modifications. This phenomenon, also termed reactive astrocyte or astrogliosis, results in loss of brain homeostatic functions and leads to the occurrence of neurological and neuropsychiatric disorders (*Verkhratsky et al., 2023*). The presence of activated astrocytes which is evidenced by increased GFAP staining, was found before immune cell infiltration in MS and its animal model, EAE (*Correale and Farez, 2015*). Several lines of evidence bolster the conception that activated astrocytes are considered to be early events and contributors to lesion development in MS and EAE etiopathology (*das Neves et al., 2021*). With the accepted notion of astrocyte contributions to MS or EAE, mounting interest has been focused on dissecting how astrocytes are reactive.

Activated immune cells, like cancer cells, require higher biosynthetic and energy needs for immune response, proliferation, and survival. This involves reprogramming of their metabolic pathways. Proinflammatory immune cells, including reactive astrocytes, usually undergo a metabolic switch from oxidative phosphorylation to Warburg-type glucose metabolism (*Vaupel and Multhoff, 2021*; *Xiong et al., 2022*). Moreover, elevated level of aerobic glycolysis has been characterized in astrocyte of MS patients (*Afzal et al., 2020*; *Nijland et al., 2015*). Elevated glycolysis is crucial for sustaining astrocyte proliferation, the secretion of proinflammatory cytokines and neurotrophic factors and subsequent neuronal loss in the CNS. As such, deciphering glycolysis-dominant metabolic switch in astrocytes is the basis for understanding astrogliosis and the development of neurological diseases such as multiple sclerosis (*das Neves et al., 2023*; *Xiong et al., 2022*).

Pyruvate kinase M2 (PKM2), a rate-limiting enzyme of glycolysis, is a key molecule that governs aerobic glycolysis. Low glycolytic enzyme activity of PKM2 promotes the conversion of pyruvate to lactate, which leads to aerobic glycolysis (*Lee et al., 2022*). In the cytoplasm, PKM2 exists in tetrameric form and possesses high pyruvate kinase activity. Specifically, PKM2 can translocate to the nucleus in its dimeric form. With a low-glycolytic function, nuclear PKM2 can act as a protein kinase or transcriptional coactivator to regulate proliferation, inflammation, and metabolic reprogramming of cells (*Liu et al., 2022*). The overexpression and nuclear translocation of PKM2 have been well documented in CNS disease. Moreover, nuclear PKM2 was upregulated in neutrophils and macrophages in patients with ischemic stroke (*Dhanesha et al., 2022*; *Li et al., 2022a*). Nuclear PKM2 in neurons was shown to promote neuronal loss in Alzheimer's disease (*Traxler et al., 2022*), suggesting that PKM2 is a key player in the development of neurological disease. Although previous studies have suggested that PKM2 could regulate astrocyte proliferation (*Zhang et al., 2015*), its potential function in astrocyte metabolic reprogramming and the upstream mechanisms underlying PKM2 nucleocytoplasmic shuttling are still elusive.

TRIM21 belongs to the TRIM protein family which possess the E3 ubiquitin ligase activity. In addition to its well-recognized function in antiviral responses, emerging evidences have documented the multifaceted role of TRIM21 in cell cycle regulation, inflammation, and metabolism (*Chen et al., 2022a*). Nevertheless, the precise mechanisms underlying the involvement of TRIM21 in CNS diseases remain largely unexplored.

In this report, we identified TRIM21 as the interacting protein of PKM2 and found that TRIM21 promoted the nuclear translocation of PKM2, thus contributing to astrocyte glycolysis and proliferation in EAE. Most importantly, we used the EAE model to demonstrate that targeting TRIM21-PKM2

axis alleviated the disease process. Our finding might help to understand the mechanism underlying astrocyte activation in neurological diseases and provide therapeutic target for the treatment of multiple sclerosis.

## Results

### Identification of PKM2 nuclear translocation in astrocytes during EAE

EAE is widely used as a mouse model of multiple sclerosis, which is typically induced by active immunization with different myelin-derived antigens along with adjuvants such as pertussis toxin (PTX). One widely used antigen is the myelin oligodendrocyte glycoprotein (MOG) $_{35-55}$ peptide (*Nitsch et al., 2021*), which was adopted in our current studies. To investigate whether PKM2 repositioning or aberrant expression drives astrocyte dysfunction in EAE mice, we obtained tissue samples from the spinal cords of different phases of EAE and control mice. Initially, nuclear translocation of PKM2 was observed at the onset phase, which sustained to the peak and chronic phases of the disease. Compared to the cytoplasmic localization of PKM2 in control mice, the expression level and nuclear ratio of PKM2 was elevated in different phases of EAE (*Figure 1A*, *Figure 1—figure supplement 1A and B*). Supernatant of MOG$_{35-55}$-stimulated splenocytes isolated from EAE mice were previously shown to elicit a T-cell response in the acute stage of EAE and are frequently used as an in vitro autoimmune model to investigate MS and EAE pathophysiology (*Chen et al., 2009*; *Du et al., 2019*; *Kozela et al., 2015*). To validate the expression pattern of PKM2 in astrocytes in vitro, primary astrocytes were isolated and cultured with supernatants from MOG$_{35-55}$-stimulated splenocytes (MOG$_{sup}$) of EAE. Activated astrocytes were observed following co-culture with the above-mentioned supernatant, showing obviously increased expression of GFAP, a marker of reactive astrocytes (*Figure 1B*). Consistently, compared to non-treated control astrocytes, MOG$_{sup}$-stimulated astrocytes displayed significantly higher nuclear ratio and expression levels of PKM2 (*Figure 1C and D*, *Figure 1—figure supplement 1C*). Together, these data suggest the nuclear translocation of PKM2 in astrocytes from EAE mice.

### Prevention of PKM2 nuclear transport suppresses aerobic glycolysis and proliferation in astrocytes

Metabolic switch of astrocytes to aerobic glycolysis and proliferation of astrocytes are early events in MS and EAE. To explore the contribution of PKM2 nuclear translocation to the alternation of astrocyte metabolism and function, DASA-58, the inhibitor of PKM2 nuclear transport that favors its tetramerization was used (*Palsson-McDermott et al., 2015*; *Rao et al., 2022*). Pretreatment with DASA-58 effectively reduced the nuclear ratio of PKM2 in MOG$_{sup}$ stimulated astrocytes (*Figure 2A and B*). As expected, MOG$_{sup}$ stimulation, which mimics the autoimmune response in MS patients, induced an increase in the glycolytic activity of astrocytes, as evidenced by glucose consumption and lactate production. However, these effects were significantly counteracted by DASA-58 treatment (*Figure 2C*). To further confirm these result, glycolysis-related enzymes and transcription factors including LDHA, PKM2 and c-myc were examined. Among these proteins, DASA-58 pretreatment significantly inhibited the upregulation of phosphorylated c-myc triggered by MOG$_{sup}$ stimulation, without any notable effect on the total level of PKM2 (*Figure 2D*, *Figure 2—figure supplement 1*).

To determine whether DASA-58 could alter astrocyte proliferation, CCK-8 and EdU assays were performed. *Figure 2E* showed that treatment with 25 µM and 50 µM DASA-58 impaired the proliferation of astrocytes, and 50 µM owned better effect. Additionally, EdU incorporation assays showed that 50 µM DASA-58 mostly abrogated the MOG$_{sup}$-induced astrocyte proliferation (*Figure 2F and G*). In addition, DASA-58 pretreatment reduced the expression of inflammatory cytokines including *IL-6, TNF-α, and iNOS* in MOG$_{sup}$-stimulated astrocytes (*Figure 2—figure supplement 2*). From the above results, we can conclude that abrogation of PKM2 nuclear transport can markedly decrease the proliferation and glycolysis of astrocytes.

### Nuclear PKM2 promotes the activation of NF-κB and STAT3 pathways

Upon nuclear translocation, PKM2 acquires protein kinase and transcriptional coactivator activities. As nuclear PKM2 has been reported to interact with STAT3 and NF-κB, which are dominant signaling pathways involved in orchestrating cell proliferation, inflammation and glycolysis, we were curious to investigate whether nuclear PKM2 regulates the activation of these two pathways. The activation of

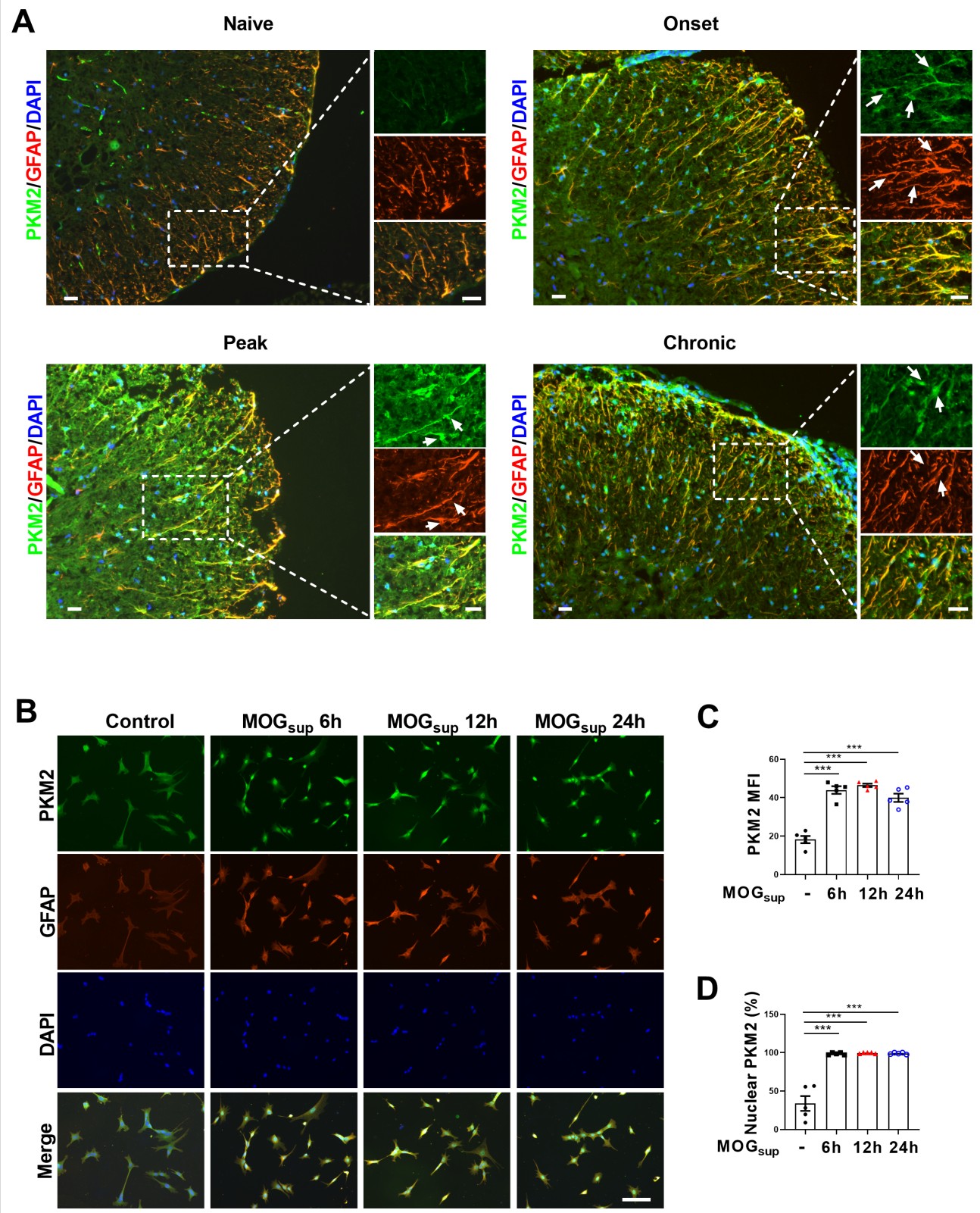

**Figure 1.** Nuclear translocation of PKM2 in astrocytes of EAE mice. (**A**) Immunofluorescence staining of PKM2 with GFAP (astrocyte marker) in spinal cord of control mice (n=4) and MOG$_{35-55}$-induced EAE mice. Disease onset (dpi 7–17, n=3), peak (dpi 14–24, n=4) and chronic (dpi 21–26, n=2) were defined dependent on the EAE course. Scale bar: 20 μm. White arrows indicated nuclear PKM2. (**B**) Immunofluorescence staining of PKM2 (green) with GFAP (red) in non-treated primary astrocytes (control) or primary astrocytes cultured with splenocytes supernatants of MOG35–55-induced EAE mice

*Figure 1 continued on next page*

*Figure 1 continued*

(MOG$_{sup}$) for different time points (6 hr, 12 hr, and 24 hr). DAPI (blue) was used as a nuclear staining. Scale bar: 100 μm. (**C**) Mean fluorescence intensity (MFI) of PKM2 in different groups of **B** was calculated by ImageJ. (**D**) Nuclear PKM2 ratio in different groups of (**B**) were calculated. Five fields of views per group were included in the analysis. The number of nuclear PKM2 was quantified by Image-Pro Plus software manually (eg. nuclear or cytoplasmic based on DAPI blue staining). The proportion of nuclear PKM2 is determined by normalizing the count of nuclear PKM2 to the count of nuclear DAPI, which represents the number of cell nuclei. Data are represented as mean ± SEM, one-way ANOVA. ***$p<0.001$. SEM, standard error of the mean.

The online version of this article includes the following source data and figure supplement(s) for figure 1:

**Source data 1.** Source data for *Figure 1C and D*.

**Figure supplement 1.** Quantification of nuclear ratio of PKM2 in astrocytes and mean fluorescence intensity of PKM2 in control and EAE mice.

**Figure supplement 1—source data 1.** Source data for *Figure 1—figure supplement 1*.

STAT3 and NF-κB requires two critical steps: phosphorylation of key components, nuclear translocation and retention of STAT3 or p65/p50 subunits. MOGsup stimulation increased the phosphorylation of STAT3 and NF-κB pathways. As expected, DASA-58 pretreatment partially attenuated the phosphorylation of STAT3 and NF-κB pathways following MOG$_{sup}$ stimulation (*Figure 3A–C*). DASA-58 also affected the activation state of astrocytes, as evidenced by the reduced expression of GFAP (*Figure 3—figure supplement 1A*). Phosphorylation only contributes to the transient activation of STAT3 and NF-κB, and constant activation also requires the nuclear retention of STAT3 and p50/p65. To test our hypothesis that nuclear PKM2 might promote the retention of p50/p65 and STAT3, we purified nuclear and cytoplasmic proteins. Western blotting assays showed that inhibiting PKM2 nuclear localization with DASA-58 suppressed the nuclear retention of p50/p65 and STAT3 (*Figure 3D*, *Figure 3—figure supplement 1B, C*).

To further elucidate the underlying mechanism of how PKM2 regulated the nuclear retention of STAT3 and NF-κB, immunoprecipitation experiments were carried out in primary astrocytes. The results confirmed the endogenous binding between PKM2 and NF-κB subunits p50/65, as well as between PKM2 and STAT3 (*Figure 3E*). Therefore, nuclear PKM2 interacts with p50/p65 and STAT3, favoring their nuclear retention and the sustained activation of NF-κB and STAT3 signaling pathways.

## E3 ligase TRIM21 interacts with PKM2 in astrocytes

With deepening of the research, amounting evidences support that post-translational modifications (PTMs), representing by ubiquitination, acetylation, sumoylation and phosphorylation are major mechanisms to regulate the process of PKM2 nuclear translocation. To illustrate underlying mechanism accounting for nuclear translocation of PKM2 in astrocytes, we sought to identify PKM2-interacting proteins. Here, unstimulated and MOG$_{sup}$-stimulated primary astrocytes were subjected to PKM2 immunoprecipitation, followed by mass spectrometry. Several enzymes involved in glycolysis and gluconeogenesis including ENO1, ALDOA, MDH2, LDHA, and LDHC were identified to be interacted with PKM2 (*Figure 4A*, *Figure 4—figure supplement 1A*). Analysis of biological processes according to Gene Ontology (GO) terms confirmed that the binding proteins of PKM2 are enriched in metabolic processes (*Figure 4B*, *Figure 4—figure supplement 1B*). Moreover, the results of KEGG and Wikipathway enrichment analysis indicate that PKM2-interacting proteins were enriched in glycolysis, gluconeogenesis and NF-κB pathway (*Figure 4C and D*, *Figure 4—figure supplement 1C, D*). Amongst these potential interacting proteins, the most attracting one is TRIM21, an E3 ligase involved in the process of ubiquitination (*Figure 4A*). Coincidentally, we previously reported the proinflammatory role of TRIM21 in keratinocytes by ubiquitylating the p50/p65 subunits of NF-κB (*Yang et al., 2021*). We were curious to verify whether TRIM21 interacted with and regulated the subcellular localization of PKM2 in astrocytes. Molecular docking revealed a strong binding affinity between PKM2 and TRIM21 (*Figure 4E*, left). TRIM21 is predicted to bound to PKM2 via hydrogen bonds between the amino acids of the two molecules (*Figure 4E*, right). By immunoprecipitation assays, we demonstrated the endogenous binding of PKM2 with TRIM21 in primary astrocytes (*Figure 4F*). To further confirm the results of PKM2-TRIM21 interaction, plasmids of Myc-tagged TRIM21 and Flag-tagged PKM2 were constructed. Reciprocal co-immunoprecipitation with either Myc or Flag antibodies verified exogenous binding between PKM2 and TRIM21 (*Figure 4G and H*). To map the binding domains between PKM2 and TRIM21, a series of truncation with deletion (Δ) of various domains of TRIM21 and PKM2 were constructed. The deletion of C-terminal PRY-SPRY domain abolished the binding between

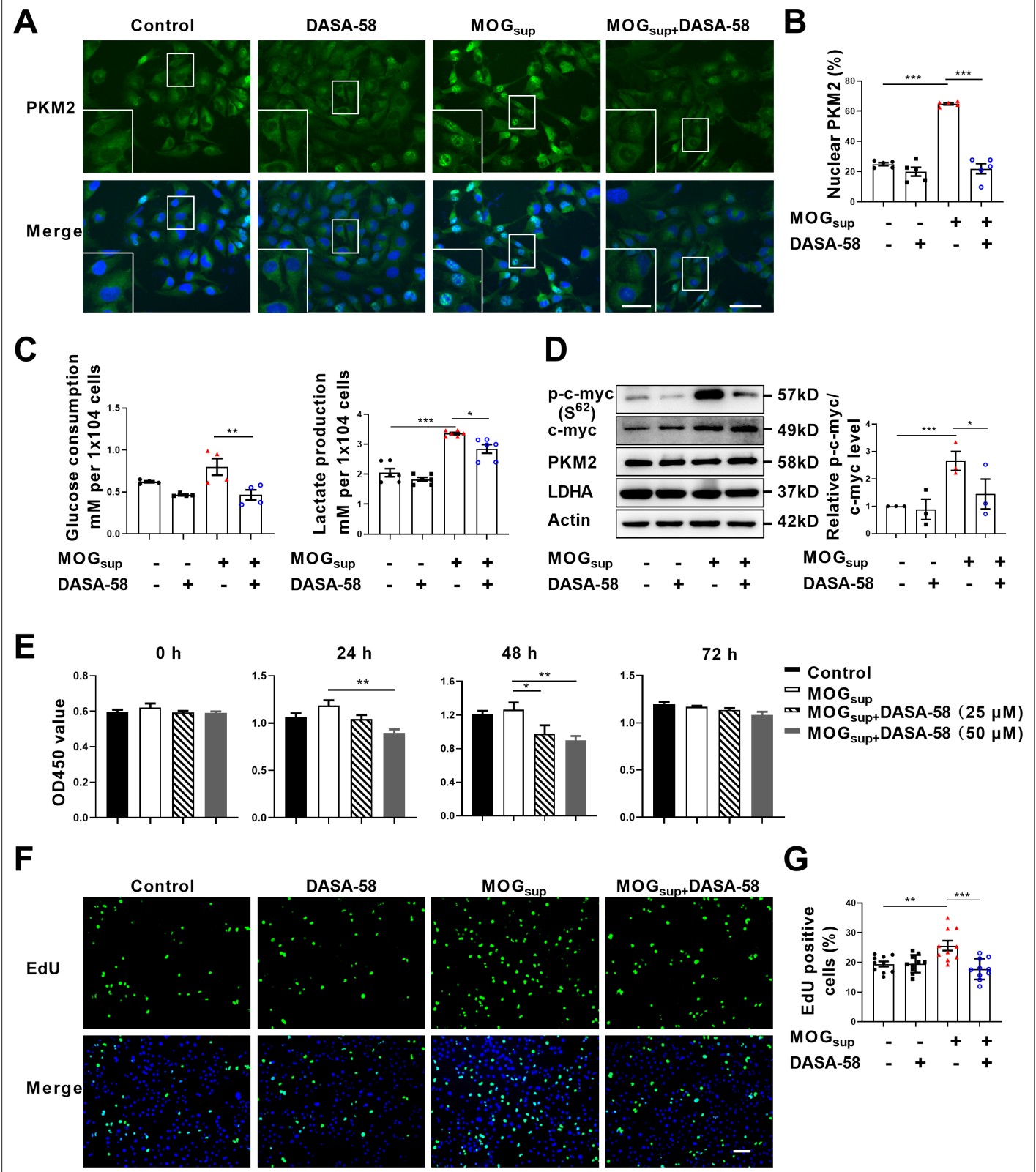

**Figure 2.** Prevention of PKM2 nuclear transport reduced the glycolysis and proliferation of primary astrocytes. (**A**) Verification of DASA-58 effect on the inhibition of PKM2 nuclear transport by immunofluorescence. Primary astrocytes were pretreated with 50 μM DASA-58 for 30 min and stimulated with MOG$_{sup}$ for 12 hr. Scale bar: 50 μm. Scale bar in enlarged image: 20 μm. (**B**) Nuclear ratio of PKM2 in each group was calculated. Five fields of views per group were included in the analysis. (**C**) Glycolysis level of astrocytes in each group was assessed by lactate production (N=5) and glucose consumption

*Figure 2 continued on next page*

*Figure 2 continued*

(N=4) assays. (**D**) Effect of DASA-58 on protein levels of glycolytic enzymes p-c-myc, LDHA and PKM2 were measured by western blotting. Right panel shows the quantification of phospho-c-myc expression normalized to total c-myc levels. (**E**) Proliferation of astrocytes were measured by CCK8. N=5. (**F**) Proliferation of astrocytes were measured by EdU assays. (**G**) EdU-positive cells in each group was calculated from ten fields of views per group. Scale bar: 100 μm. The blot is representative of three independent experiments. Data are represented as mean ± SEM, one-way ANOVA. *p<0.05; **p<0.01; ***p<0.001.

The online version of this article includes the following source data and figure supplement(s) for figure 2:

**Source data 1.** Source data for *Figure 2B–E and G*.

**Source data 2.** Uncropped and labeled gels for *Figure 2D*.

**Source data 3.** Raw unedited gels for *Figure 2D*.

**Figure supplement 1.** Quantification of PKM2 protein level in astrocytes treated with MOG$_{sup}$ or MOG$_{sup}$ pretreated with DASA-58.

**Figure supplement 1—source data 1.** Source data for *Figure 2—figure supplement 1*.

**Figure supplement 2.** qPCR analysis of mRNA levels of inflammatory cytokines.

**Figure supplement 2—source data 1.** Source data for *Figure 2—figure supplement 2*.

TRIM21 to PKM2, which indicated that PRY-SPRY domain of TRIM21 was responsible for the interaction with PKM2 (*Figure 4I*). However, the deletion of either N- or C-terminal of PKM2 did not affect the binding between TRIM21 to PKM2, indicating that AB domain (44–388 amino acids) of PKM2 might interact with TRIM21 (*Figure 4J*).

## Upregulated TRIM21 expression in astrocytes of EAE mice and in activated primary astrocytes

TRIM21 is a multifunctional E3 ubiquitin ligase that plays a crucial role in orchestrating diverse biological processes, including cell proliferation, antiviral responses, cell metabolism and inflammatory processes (*Chen et al., 2022a*). Moreover, our previous study is the first to uncover the upregulation of TRIM21 in the epidermis of psoriatic patients, an autoimmune skin disease characterized by hyperproliferation of epidermal keratinocytes (*Yang et al., 2018*; *Yang et al., 2021*). To determine the relative expression of TRIM21 in astrocytes of EAE mice, we firstly performed single-cell RNA sequencing (scRNA-seq) on brain samples from the control, EAE peak and chronic stages. ScRNA-seq analysis revealed differential expression of TRIM21 in multiple cell populations. Compared to that in other cell types, *TRIM21* expression in astrocytes was relatively high (*Figure 5A*). We identified 12 astrocyte subpopulations, whereas TRIM21 expression was divergent in different astrocyte clusters. Most importantly, TRIM21 expression was augmented in astrocytes in both peak and chronic phases of EAE compared to that in control mice (*Figure 5B–D*). Consistently, bioinformatic analysis of the GEO database (GSE136358) revealed significant elevation of TRIM21 expression in astrocytes at the onset, peak, and chronic phases of EAE disease (*Figure 5E*).

To further confirm the results of TRIM21 expression from scRNA-seq and GEO datasets, activated astrocytes were mimicked by stimulating primary astrocytes with MOG$_{sup}$. Compared to those in non-stimulated astrocytes, qPCR and western blotting analysis revealed dramatic increases in *TRIM21* mRNA and protein expression in activated astrocytes (*Figure 5F and G*). Moreover, immunofluorescence staining further demonstrated that TRIM21 expression was greater in astrocytes from EAE mice when compared with control mice (*Figure 5H*). Taken together, our results uncover the upregulated expression of TRIM21 in astrocytes of EAE mice, which imply that the ectopic expression of this ubiquitin ligase TRIM21 might be a potent regulator of PKM2 repositioning in the nucleus.

## TRIM21 promotes ubiquitylation and the nuclear translocation of PKM2

Ubiquitination is endowed with multifaceted function to regulate degradation, localization and activation of substrate proteins. As PKM2 has been demonstrated to be the interacting protein and substrate of TRIM21, we next examined the impact of TRIM21 on PKM2 localization. Overexpression of TRIM21 induced a robust increase in the nuclear ratio of PKM2 (*Figure 6A*, *Figure 6—figure supplement 1A*). In contrast, knockdown of TRIM21 led to a reduction in the nuclear ratio of PKM2 (*Figure 6B*, *Figure 6—figure supplement 1B*). To a greater extent, TRIM21 was found to be a potent driver of PKM2 translocation in astrocytes of EAE. To deeply unveil the mechanism of TRIM21-mediated

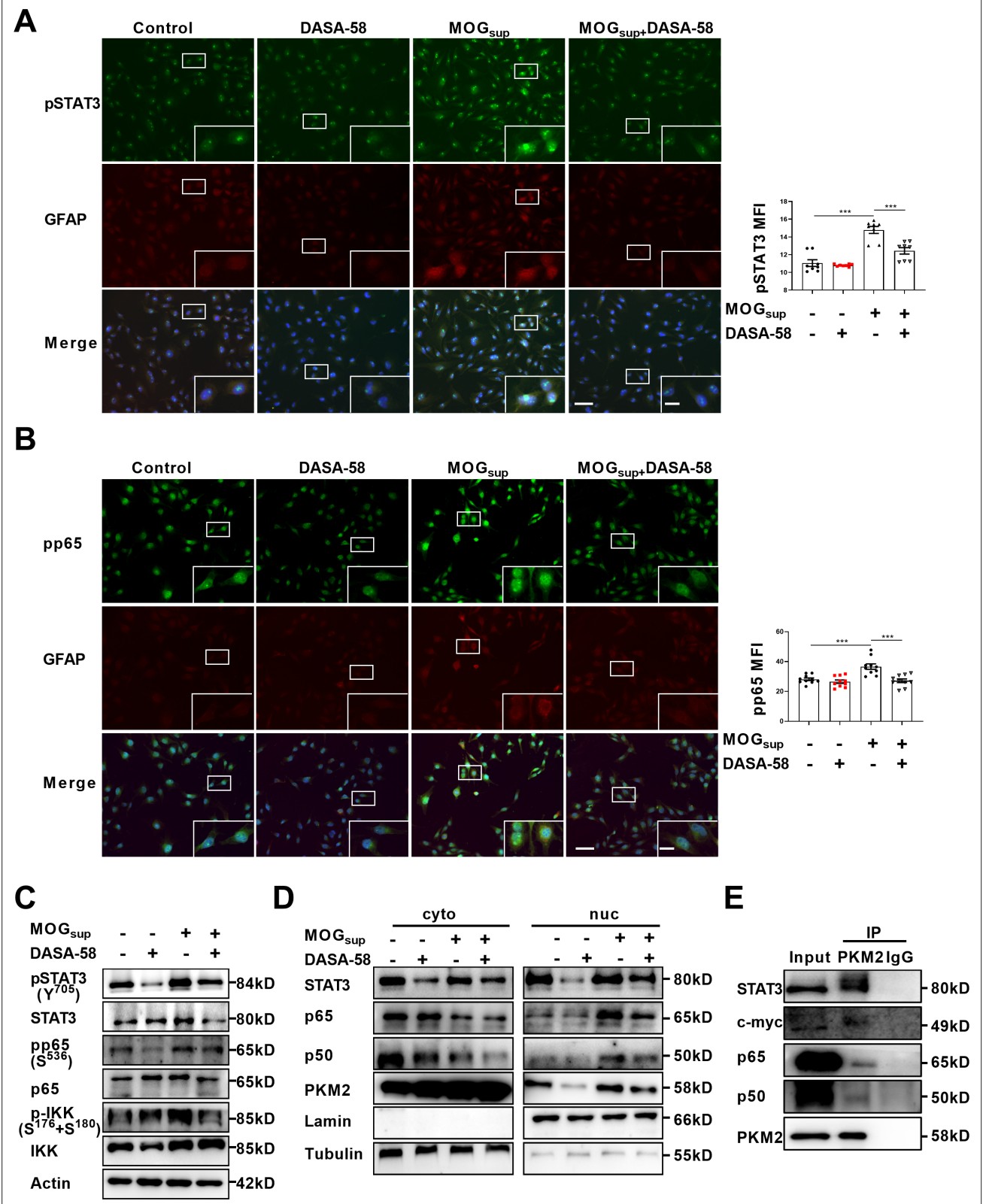

**Figure 3.** PKM2 interacted with STAT3 and NF-$\kappa$B and promoted their activation in astrocytes. (**A–B**) Immunofluorescence staining of phospho-STAT3 (**A**) or phospho-p65 (**B**) with GFAP in astrocytes. Primary astrocytes were pretreated with 50 µM DASA-58 for 30 min and stimulated with MOG$_{sup}$ for 12 hr. Scale bar: 100 µm. Scale bar in enlarged image: 20 µm. (**C**) Western blotting analysis showed that DASA-58 inhibited the activation of NF-$\kappa$B and STAT3 induced by MOG$_{sup}$ stimulation. (**D**) Nuclear-cytoplasmic protein extraction analysis showed cytoplasmic and nuclear protein levels STAT3 and

*Figure 3 continued on next page*

*Figure 3 continued*

p50/p65 upon DASA-58 treatment. (**E**) Immunoprecipitation demonstrated the interaction between PKM2 and STAT3, c-myc and p50/p65 subunits of NF-$\kappa$ B in primary astrocyte. Data are represented as mean ± SEM, one-way ANOVA. **p<0.01; ***p<0.001. SEM, standard error of the mean.

The online version of this article includes the following source data and figure supplement(s) for figure 3:

**Source data 1.** Source data for *Figure 3A and B*.

**Source data 2.** Uncropped and labeled gels for *Figure 3C*.

**Source data 3.** Raw unedited gels for *Figure 3C*.

**Source data 4.** Uncropped and labeled gels for *Figure 3D*.

**Source data 5.** Raw unedited gels for *Figure 3D*.

**Source data 6.** Uncropped and labeled gels for *Figure 3E*.

**Source data 7.** Raw unedited gels for *Figure 3E*.

**Figure supplement 1.** Quantification results of GFAP and cyto-nuclear protein levels in astrocytes treated with MOG$_{sup}$ or MOG$_{sup}$ pretreated with DASA-58.

**Figure supplement 1—source data 1.** Source data for *Figure 3—figure supplement 1*.

binding with PKM2, the ubiquitination linkage type was investigated. In addition to K48-linked ubiquitination, which directs proteins for degradation, K63-linked ubiquitination is implicated in the regulation of protein localization and activation. Immunoprecipitation implied that K63-linked ubiquitination of PKM2 was enhanced upon overexpression of TRIM21 (*Figure 6C*). Collectively, the data showed that TRIM21 promoted K63-linked ubiquitination of PKM2 and facilitated its nuclear translocation in astrocytes.

## TRIM21 promotes aerobic glycolysis and proliferation by enhancing PKM2 nuclear function in astrocytes

As TRIM21 promoted the nuclear translocation of PKM2, we explored the impact of TRIM21 on the nuclear function of PKM2. Our results showed that the levels of phosphorylated STAT3 and p65 were significantly increased upon TRIM21 overexpression (*Figure 6D*). We next examined whether TRIM21 could affect the binding of PKM2 to c-myc, STAT3 and NF-κB subunits. As shown in *Figure 6E*, overexpression of TRIM21 promoted the binding of PKM2 to c-myc, STAT3 and p50 subunit of NF-κB. Nuclear PKM2 contributed to nuclear retention of STAT3 and NF-κB, which retained the constant activation of these two signaling pathways. We were curious to investigate whether TRIM21 is involved in this process. Notably, fractionation analysis revealed that overexpression of TRIM21 increased the nuclear accumulation of c-myc, STAT3 and p50/p65 subunits. Conversely, pretreatment with DASA-58, which abrogated the nuclear translocation of PKM2, diminished the nuclear retention of the aforementioned transcription factors (*Figure 6F*). These findings revealed that the TRIM21-mediated nuclear translocation of PKM2 promoted its nuclear function.

To further assess the functional consequences of TRIM21-mediated nuclear translocation of PKM2, the glycolytic activity and proliferation of astrocytes were measured. As shown in *Figure 6G*, TRIM21 overexpression increased the ratio of EdU positive cells. However, the increase in astrocyte proliferation caused by TRIM21 upregulation was significantly antagonized by the DASA-58 treatment (*Figure 6G*). Similarly, upregulated TRIM21 promoted lactate production and glucose consumption, which were reversed by DASA-58 (*Figure 6H*). In summary, our results indicate that nuclear PKM2-mediated metabolic reprogramming is crucial for TRIM21-stimulated proliferation of astrocytes.

## TRIM21 knockdown in astrocyte or TEPP-46 treatment inhibits the development of EAE

To determine the therapeutic effect of TRIM21 knockdown in astrocytes on EAE, sh*TRIM21* and control lentivirus were given to mice by intracerebroventricular administration at disease onset (15 days post immunization). As expected, shTRIM21 treatment suppressed disease severity of EAE. At the end time point at day 22 p.i., shTRIM21-treated group showed reduced disease scores compared to control groups (*Figure 7A*). To validate the impact of TRIM21 knockdown in astrocytes on the nuclear translocation of PKM2 in vivo, immunostaining of PKM2 was performed. Consistent with in vitro results, nuclear PKM2 in astrocytes was reduced in shTRIM21-treated group when compared to shVec-treated

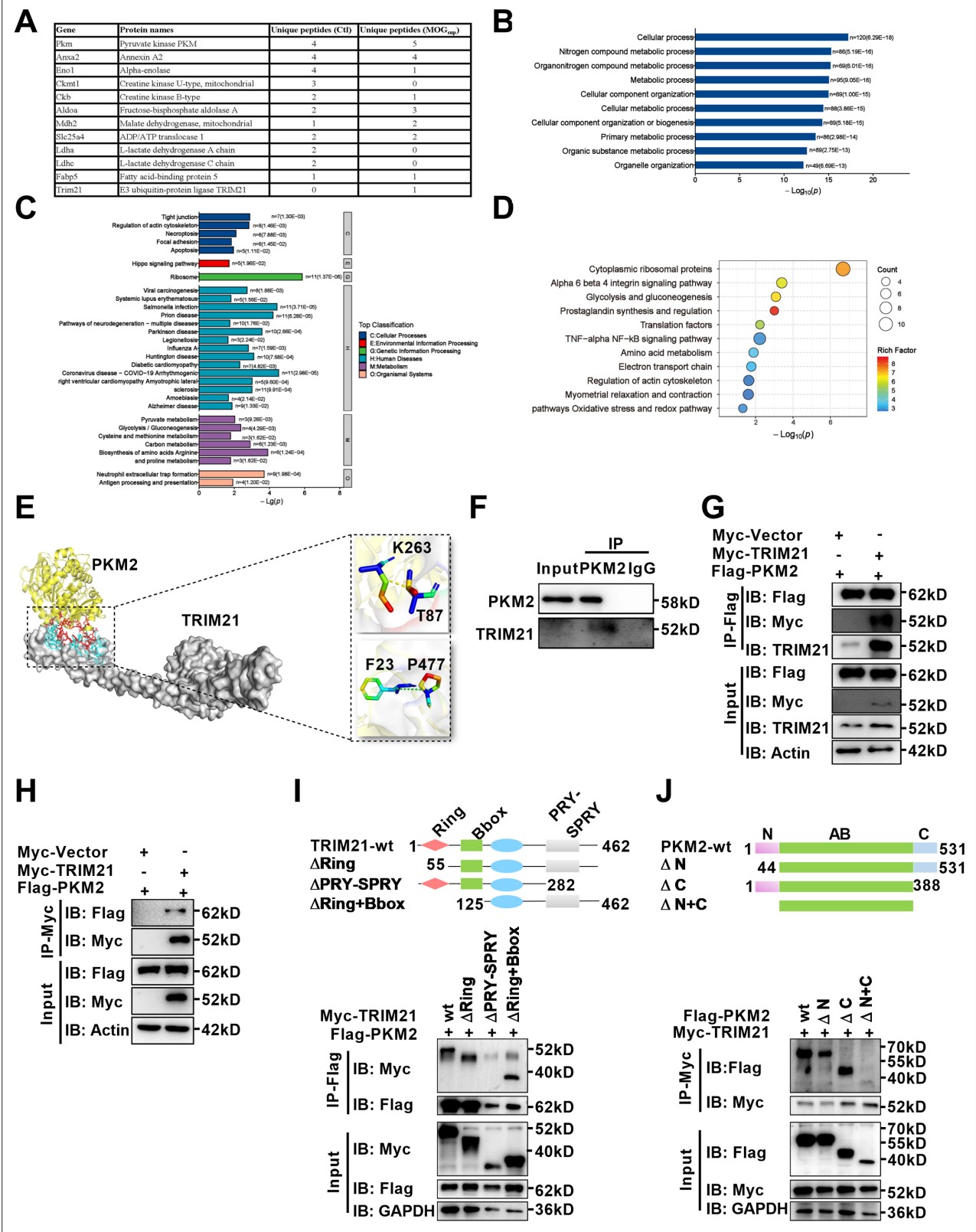

**Figure 4.** Identification of interaction between E3 ligase TRIM21 and PKM2 in astrocytes. (**A**) Mass spectrometry (MS) showed the list of metabolic-related proteins that potentially interact with PKM2 in unstimulated (Ctl) and MOG$_{sup}$-stimulated primary astrocytes. TRIM21 was identified to interact with PKM2. (**B–D**) Biological process of GO term (**B**), KEGG pathway (**C**) and Wikipathway (**D**) analysis of proteins identified by MS. (**E**) Interaction between PKM2 and TRIM21 was predicted with molecular docking and showed by PyMol. The hydrogen bonds were formed between Phe23, Thr87 of

*Figure 4 continued on next page*

*Figure 4 continued*

TRIM21 and Pro477, Lys 263 of PKM2. (**F**) Immunoprecipitation showed the interaction between endogenous PKM2 and TRIM21 in primary astrocyte. (**G–H**) Primary astrocytes were transfected with Myc-tagged TRIM21 and Flag-tagged PKM2, immunoprecipitation with anti-Flag (**G**) or anti-Myc (**H**) showed the exogenous binding between PKM2 and TRIM21 in astrocytes. (**I**) Full-length TRIM21 and a series of TRIM21 mutants with deletion (Δ) of various domains (top panel). 293 T cells were co-transfected with Flag-PKM2 and WT Myc-TRIM21 or their truncation mutants for 48 h. Immunoprecipitation was performed. (**J**) Full-length PKM2 and a series of PKM2 mutants with deletion (Δ) of various domains (top panel). 293 T cells were co-transfected with Myc-TRIM21 and WT Flag-PKM2 or their truncation mutants for 48 hr. Immunoprecipitation was performed.

The online version of this article includes the following source data and figure supplement(s) for figure 4:

**Source data 1.** Uncropped and labeled gels for *Figure 4F*.

**Source data 2.** Raw unedited gels for *Figure 4F*.

**Source data 3.** Uncropped and labeled gels for *Figure 4G*.

**Source data 4.** Raw unedited gels for *Figure 4G*.

**Source data 5.** Uncropped and labeled gels for *Figure 4H*.

**Source data 6.** Raw unedited gels for *Figure 4H*.

**Source data 7.** Uncropped and labeled gels for *Figure 4I*.

**Source data 8.** Raw unedited gels for *Figure 4I*.

**Source data 9.** Uncropped and labeled gels for *Figure 4J*.

**Source data 10.** Raw unedited gels for *Figure 4J*.

**Figure supplement 1.** Mass spectrometry results of PKM2-interacting proteins in astrocytes.

group (*Figure 7—figure supplement 1A*). To further measure the effect of TRIM21 knockdown in astrocytes on pathological changes in EAE mice, HE and LFB staining were performed. As expected, inflammation and demyelination were less pronounced in shTRIM21-treated group (*Figure 7B and C*). Staining for TRIM21 showed that TRIM21 expression was reduced in astrocytes after intracerebro-ventricular injection of shTRIM21 lentivirus (*Figure 7D*). Demyelination lesions were also evaluated by myelin basic protein (MBP) staining. Knockdown of TRIM21 in astrocytes significantly increased MBP positive areas, which indicated the inhibited demyelination in shTRIM21-treated group compared with control group (*Figure 7E*). In EAE, microglia and astrocyte activation are linked with demyelination, we next stained GFAP and IBA1 to measure the activation of astrocytes and microglia. Knocking down TRIM21 in astrocytes decreased GFAP expression on spinal cord sections. The decrease of GFAP+ cell numbers was observed in both gray and white matter from shTRIM21-treated mice (*Figure 7F*). For activated microglia expressing IBA1, similar results were observed. Control group showed a wide-spread activation, while shTRIM21-treated group showed a significant decrease in IBA1 positive cells in both white matter and gray matter of spinal cord (*Figure 7G*).

Therapeutic potential of PKM2 nuclear translocation inhibition with TEPP-46 was also tested in the EAE model. TEPP-46 is a selective allosteric activator for PKM2, showing little or no effect on other pyruvate isoforms. It promotes the tetramerization of PKM2, thereby diminishing its nuclear translocation (*Anastasiou et al., 2012*; *Angiari et al., 2020*). To test the effect of TEPP-46 on the development of EAE, the 'preventive treatment' (i.e. administration before disease onset) was administered. Intraperitoneal treatment with TEPP-46 at a dosage of 50 mg/kg every other day from day 0 to day 8 post-immunization with $MOG_{35-55}$ resulted in decreased disease severity (*Figure 7—figure supplement 2A*). The in vivo effect of TEPP-46 in inhibiting the nuclear translocation of PKM2 in astrocytes were also confirmed (*Figure 7—figure supplement 1B*). TEPP-46-treated mice exhibited reduced inflammation and demyelination (*Figure 7—figure supplement 2B–E*). The activation of GFAP positive astrocytes and IBA1 positive microglia were correspondingly reduced in TEPP-46-treated mice (*Figure 7—figure supplement 2D and E*). Taken together, these results showed that TRIM21 deficiency in astrocytes or prevention of PKM2 nuclear translocation substantially inhibited inflammation and myelin depletion in EAE mice.

## Discussion

Reactive astrocytes, or astrocyte activation are recognized as common features of CNS pathology, including neurodegenerative and demyelinating diseases (*Patani et al., 2023*). Preferential metabolic

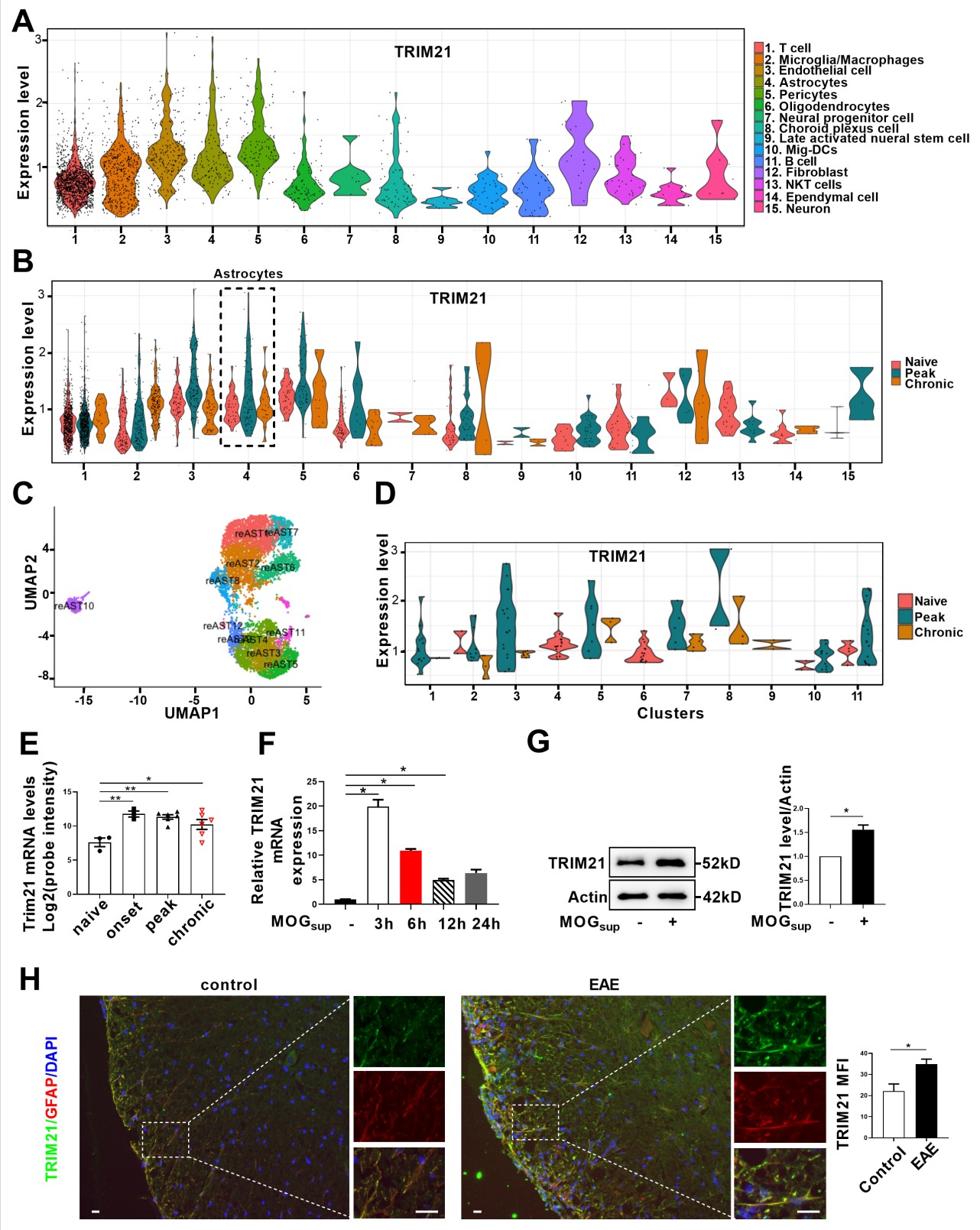

**Figure 5.** TRIM21 expression is upregulated in astrocytes of EAE mice. (**A–D**) Single-cell RNA-seq profiles from naive and EAE mice (peak and chronic phase) CNS tissues. Naive (n=2); peak (dpi 14–24, n=3); chronic (dpi 21–26, n=2). (**A**) Violin plots displaying the expression of TRIM21 across the cell types identified. (**B**) Violin plots displaying the expression of TRIM21 in different phases of EAE and naive mice across the cell types identified. Expression of TRIM21 was shown to be elevated in astrocytes of EAE mice (peak and chronic) compared with naive mice. (**C**) UMAP representation of

*Figure 5 continued on next page*

*Figure 5 continued*

12 clusters generated from sub-clustering of astrocytes. (**D**) Violin plots displaying the expression of TRIM21 at peak, chronic phases from EAE and naive mice in subclusters of astrocytes. (**E**) Analysis of TRIM21 mRNA expression in astrocytes from spinal cord during three stages (onset, peak, and chronic) of EAE and naive mice from GEO dataset GSE136358 (one-way ANOVA, *p<0.05; **p<0.01). (**F**) Primary astrocytes were treated with or without MOG$_{sup}$ for different time points. Analysis of TRIM21 expression by qPCR. (**G**) Western blotting analysis of TRIM21 protein expression in non-treated or MOG$_{sup}$-treated astrocytes. The right panel shows the quantification of TRIM21 expression normalized to β-actin loading control (Paired t test, *p<0.05). (**H**) Immunofluorescence staining showed the upregulated expression of TRIM21 in astrocytes (marker: GFAP) of EAE mice (Unpaired t test, *p<0.05). Scale bar: 20 μm. Data are represented as mean ± SEM.

The online version of this article includes the following source data for figure 5:

**Source data 1.** Source data for *Figure 5E–H*.

**Source data 2.** Uncropped and labeled gels for *Figure 5G*.

**Source data 3.** Raw unedited gels for *Figure 5G*.

switch toward aerobic glycolysis favors astrocyte transfer from 'resting' to 'reactive' state. Thus, deciphering the mechanism responsible for astrocyte metabolic switch in response to neurological disease will provide new insights and new therapeutic targets for CNS diseases. Previous studies have identified nuclear translocation of PKM2 in astrocytes following spinal cord injury (*Zhang et al., 2015*) and in a chronic inflammatory pain model (*Wei et al., 2020*), suggesting PKM2's regulatory role in aerobic glycolysis and proliferation. However, in EAE model of multiple sclerosis, whether astrocytes display PKM2 nuclear translocation and the causal mechanisms involved are still unclarified. To the best of our knowledge, this study is the first to document the nuclear translocation of PKM2 in astrocytes of EAE. Furthermore, we clarified a ubiquitination-mediated regulation of PKM2 nuclear transport. Specifically, we observed an increased expression of the E3 ubiquitin ligase TRIM21 in astrocytes from EAE mice, which promotes the nuclear translocation of PKM2, thus enhancing astrocyte activation (*Figure 8*).

Among the PTMs that regulate expression and localization of PKM2, phosphorylation is the most frequently reported type. However, it remains less understood whether ubiquitination could serve as a potential regulator. Ubiquitin-mediated degradation of PKM2 has been reported. Recently, in ovarian cancer, E3 ligase CHIP was shown to directly interact with PKM2 and mediate its degradation (*Shang et al., 2017*). TRIM family E3 ligase TRIM35 was previously shown to mediate the degradation of PKM2 in cardiomyocytes and breast cancer cells (*Lorenzana-Carrillo et al., 2022*; *Wu et al., 2022*). Although laforin/malin E3 ligase-induced ubiquitination of PKM2 did not lead to its degradation, ubiquitination in this case impaired its nuclear transport (*Viana et al., 2015*). Different from these findings, it should be emphasized that our study adds to the current knowledge that ubiquitination, in addition to SUMOylation (*Zhou et al., 2022*) and phosphorylation (*Yang et al., 2012*), could induce the relocalization of PKM2 in the nucleus.

TRIM21 is found in our study to interact with and ubiquitylate PKM2. As a traditional E3 ubiquitin ligase, multiple key molecules involved in metabolism, immunity and inflammation have been recognized as substrates of TRIM21 (*Chen et al., 2022b*). In addition to the well-known function of TRIM21 in inflammation, an increasing number of studies have suggested that TRIM21 plays a regulatory role in glucose metabolism. Glycolytic-related enzymes including PFK1 (*Tang et al., 2022*), GLUT1 (*Gu et al., 2022*) and glycolysis-related transcription factor HIF-1α (*Chen et al., 2021*) were identified to be substrates of TRIM21, and TRIM21 mediated the ubiquitin-dependent degradation of these proteins, thereby inhibiting aerobic glycolysis. Hereby, we recognized PKM2 as an interacting protein of TRIM21, and further studies are required to determine if it is a substrate of E3 ligase TRIM21. The fate of the ubiquitinated protein varies greatly, depending on the linkage type present in the ubiquitin chain. Here, we found that TRIM21 promoted K63-linked ubiquitination of PKM2, the second common type of linkage that is typically not associated with protein degradation. These findings imply that enzymes and proteins implicated in glycolysis are potential substrates of TRIM21, further suggesting that TRIM21 as a regulator of glycolysis. The limitation of the current study is the lack of mechanistic insight into the signaling pathways resulting in TRIM21 upregulation in EAE. Future studies are needed to investigate whether TRIM21 is also elevated in other CNS diseases.

The growing body of literature shows that nuclear translocation of PKM2 are related to the pathogenesis of diverse CNS diseases, such as Alzheimer's disease, ischemic stroke, glioblastoma, and spinal cord injury (*Dhanesha et al., 2022*; *Liang et al., 2016*; *Traxler et al., 2022*; *Zhang et al.,*

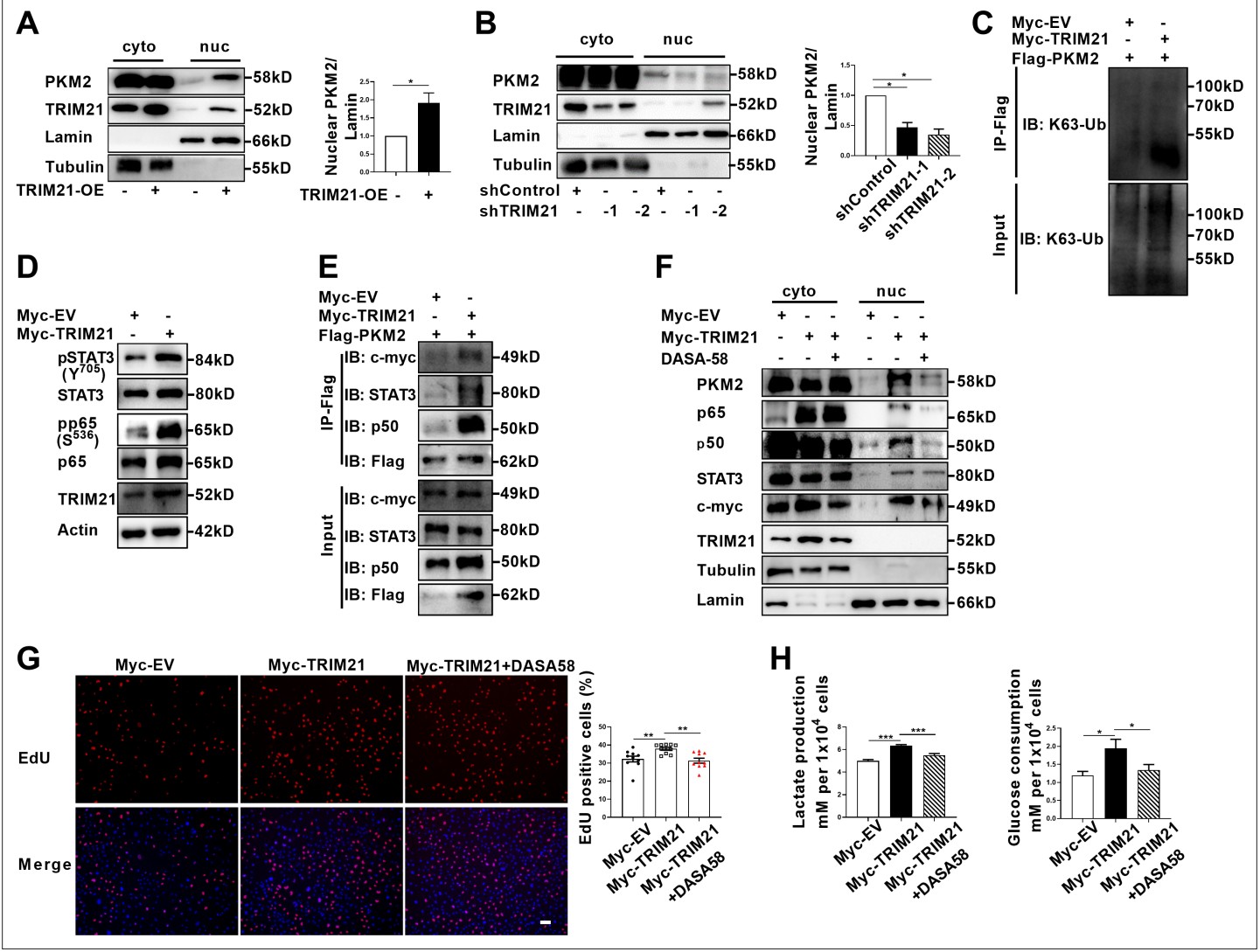

**Figure 6.** TRIM21-induced nuclear transport of PKM2 promoted glycolysis and proliferation of astrocytes. (**A**) Overexpression of TRIM21 promoted nuclear translocation of PKM2. The right panel shows the quantification of nuclear PKM2 expression level normalized to nuclear lamin level (Paired t test, *p<0.05). (**B**) TRIM21 was silenced in primary astrocytes using two independent short hairpin RNAs. Nuclear-cytoplasmic fraction analysis showed that knockdown of TRIM21 decreased nuclear ratio of PKM2. The right panel shows the quantification of nuclear PKM2 expression level normalized to nuclear lamin level (one-way ANOVA, *p<0.05). (**C**) Immunoprecipitation showed that TRIM21 promoted the K63-linked ubiquitination of PKM2. (**D**) Western blotting analysis of STAT3 and NF-κB activation in control or TRIM21-overexpressed astrocytes. (**E**) Immunoprecipitation showed that TRIM21 promoted the interaction between PKM2 and its interacting proteins c-myc, STAT3, and p50. (**F**) Prevention of PKM2 nuclear import with DASA-58 (50 μM) reduced the nuclear retention of NF-κB subunits and STAT3 in TRIM21-overexpressed astrocytes. (**G**) EdU analysis of cell proliferation in TRIM21-overexpressed, DASA-58-treated TRIM21-overexpressed cells and control astrocytes (one-way ANOVA, **p<0.01). Scale bar: 100 μm. (**H**) Glycolysis of astrocytes were measured in TRIM21-overexpressed, DASA-58 treated TRIM21-OE cells and control astrocytes (one-way ANOVA, *p<0.05; ***p<0.001). EV: empty vector. Data are represented as mean ± SEM.

The online version of this article includes the following source data and figure supplement(s) for figure 6:

**Source data 1.** Source data for *Figure 6A–B, G and H*.

**Source data 2.** Uncropped and labeled gels for *Figure 6A*.

**Source data 3.** Raw unedited gels for *Figure 6A*.

**Source data 4.** Uncropped and labeled gels for *Figure 6B*.

**Source data 5.** Raw unedited gels for *Figure 6B*.

**Source data 6.** Uncropped and labeled gels for *Figure 6C*.

**Source data 7.** Raw unedited gels for *Figure 6C*.

*Figure 6 continued on next page*

*Figure 6 continued*

**Source data 8.** Uncropped and labeled gels for *Figure 6D*.

**Source data 9.** Raw unedited gels for *Figure 6D*.

**Source data 10.** Uncropped and labeled gels for *Figure 6E*.

**Source data 11.** Raw unedited gels for *Figure 6E*.

**Source data 12.** Uncropped and labeled gels for *Figure 6F*.

**Source data 13.** Raw unedited gels for *Figure 6F*.

**Figure supplement 1.** Verification of TRIM21 overexpression and knockdown efficiency.

**Figure supplement 1—source data 1.** Uncropped and labeled gels for *Figure 6—figure supplement 1A*.

**Figure supplement 1—source data 2.** Raw unedited gels for *Figure 6—figure supplement 1A*.

**Figure supplement 1—source data 3.** Uncropped and labeled gels for *Figure 6—figure supplement 1B*.

**Figure supplement 1—source data 4.** Raw unedited gels for *Figure 6—figure supplement 1B*.

*2015*). Specifically, in ischemic stroke, upregulation of nuclear PKM2 in neutrophils has been observed in both human and mouse models. Application of ML-265 (TEPP-46) to inhibit nuclear translocation of PKM2 improved stroke outcomes (*Dhanesha et al., 2022*). In agreement with these findings, our study revealed that nuclear PKM2 levels in astrocytes are increased in different stages of EAE. Similarly, TEPP-46 treatment mitigated EAE development. Previous study demonstrated the ability of TEPP-46 in inhibiting PKM2 nuclear translocation and glycolysis in CD4$^+$ T cells, and showed its therapeutic effect in limiting inflammation in EAE (*Angiari et al., 2020*). In addition to the previous observed effect of TEPP-46 in the peripheral immune system in EAE, our study highlights the potential role of TEPP-46 in the central immune system. Glial cell activation and demyelination were inhibited following TEPP-treatment in EAE models. Our findings reveal that astrocyte PKM2 nuclear translocation contributes to astrocyte glycolysis and activation, offering insights into PKM2's pivotal role in other astrocyte-associated CNS diseases.

Although our study shed light on the role of PKM2 in astrocytes, whether PKM2 functions in a cell-specific manner or acts as a generalist warrants further studies. For example, microglia activation is a key step that contributes to CNS disorders such as multiple sclerosis and Alzheimer's disease (*Long et al., 2024*). Activated M1 microglial cells exhibit a metabolic switch toward aerobic glycolysis similar to that of astrocytes. Thus, it is highly possible that PKM2 may also be involved in microglia metabolic change and activation. As such, PKM2 might be a novel therapeutic target for the treatment of CNS disease. However, further studies are needed to decipher the role of nuclear localized PKM2 in different cells under pathological conditions to provide a thorough understanding of the biological functions of PKM2.

In the nucleus, PKM2 regulates gene expression by acting as a transcriptional co-activator or protein kinase. PKM2 is shown to interact with and phosphorylate STAT3 in cancer cells, immune cells and keratinocytes (Chen C. et al., 2022, *Yang et al., 2023*; *Zhou et al., 2022*). In addition to regulating cell proliferation and T cell differentiation, STAT3 directly and indirectly modulates genes linked to glucose metabolism, particularly glycolysis. STAT3 positively regulates the expression of HIF-1α and MYC, two crucial promoters of Warburg effect (*Tošić and Frank, 2021*). Moreover, STAT3 directly regulates the glycolysis-related genes by binding to the promoter of *Slc2a1* (*Li et al., 2022b*). Aligning with these findings, our results reveal that PKM2 interacts with and phosphorylates STAT3, thereby providing a rationale for the proliferation and enhanced glycolysis observed in astrocytes of EAE. Additionally, we also identified other transcriptional factors, such as p65-NF-κB, that interact with PKM2. Given the crucial role of NF-κB in inflammation, further research is needed to determine if PKM2-p65 is involved in astrocyte inflammation.

In addition to identifying the contribution of TRIM21 to PKM2 nuclear translocation and TRIM21-PKM2 axis in promoting astrocyte glycolysis and proliferation, the therapeutic effect of TRIM21 in EAE was also tested. By using lentivirus with astrocyte-specific GFAP promoter, the knockdown of TRIM21 in astrocytes has been successfully achieved. This approach by using lentivirus to deliver shRNA into astrocytes has been previously reported by our group, in which shAct1 lentivirus showed potency for the treatment of EAE (*Yan et al., 2012*). In the presented study, we showed that blocking TRIM21 pathway effectively ameliorated disease severity of EAE, which is evidenced by

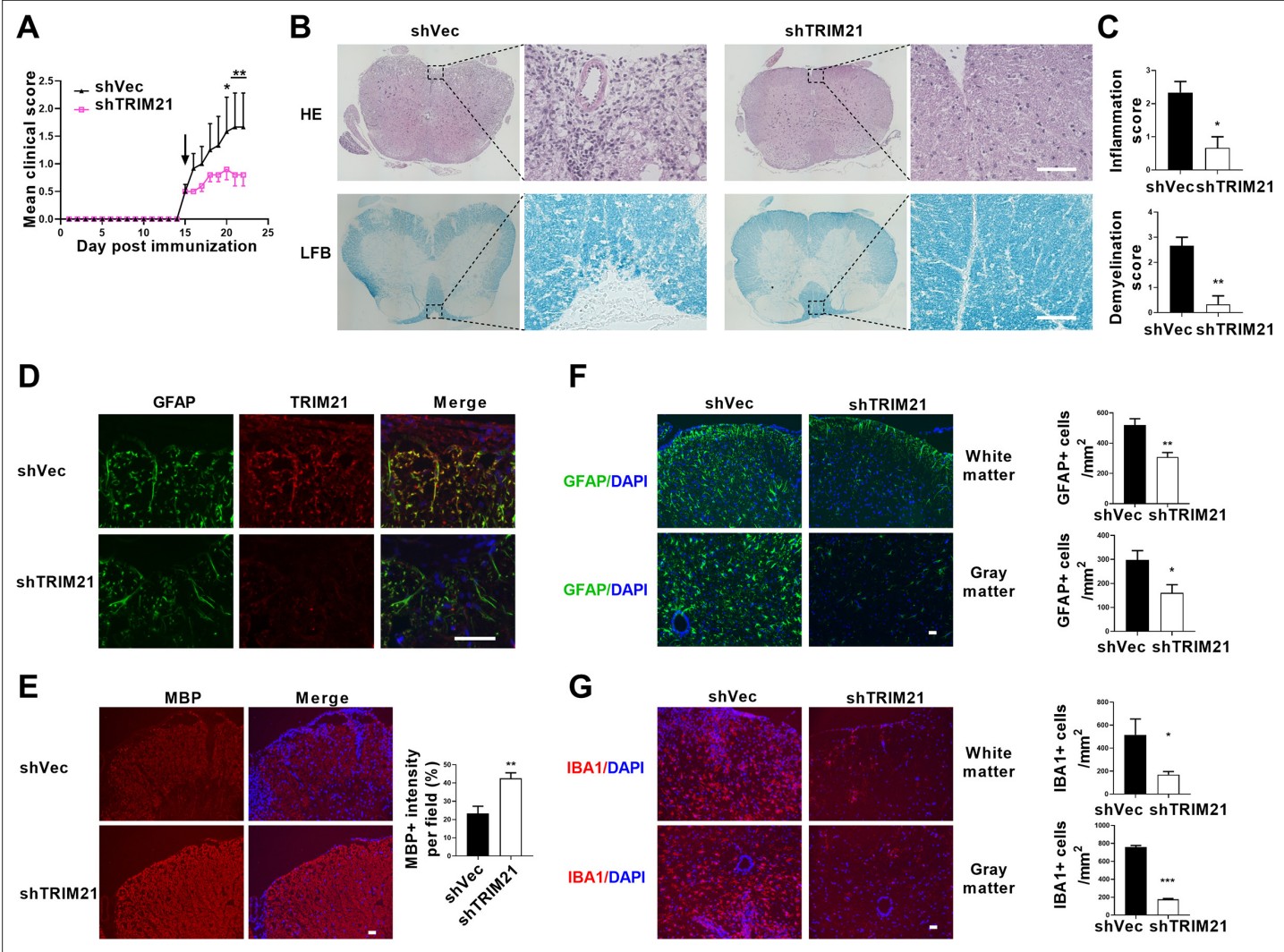

**Figure 7.** Intracerebroventricular injection of shTRIM21 ameliorates disease severity of Experimental Autoimmune Encephalomyelitis (EAE). C57BL/6 mice were injected i.c.v with 1x10⁷ IU shTRIM21 or control lentivirus (shVec) 15 days p.i. (onset). Mice were sacrificed at day 22 p.i. and spinal cords were harvested. (**A**) Disease was scored daily on a 0–5 scale. N=5–6 mice in each group. (**B**) Spinal cord sections were stained for markers of inflammation by hematoxylin and eosin (H&E) and demyelination by Luxol fast blue (LFB), respectively. (**C**) Scoring of inflammation (H&E) and demyelination (LFB) on a 0–3 scale. (**D**) TRIM21 expression in spinal cord of mice from shVec and shTRIM21 group was measured by immunofluorescence. (**E**) Demyelination in each group was assessed by MBP staining. MBP intensity was measured in the white matter of the spinal cord using Image-Pro. (**F–G**) Immunostaining of GFAP (**F**) and IBA1 (**G**) on spinal cord sections of shVec and shTRIM21-treated EAE mice. White matter and gray matter are shown as representative images. Quantification of GFAP positive cells/mm², IBA1 positive cells/mm² in both the white matter and gray matter. The measured areas included 3–5 fields per group. i.c.v., intracerebroventricular; p.i., postimmunization. Scale bar: 50 μm. Data are represented as mean ± SEM. *p<0.05; **p<0.01; ***p<0.001, as determined by two-way ANOVA analysis (**A**) or unpaired Student's t test (**C, F–G**).

The online version of this article includes the following source data and figure supplement(s) for figure 7:

**Source data 1.** Source data for *Figure 7A, C and E–G*.

**Figure supplement 1.** PKM2 expression and localization in shTRIM21-treated and TEPP-treated EAE mice.

**Figure supplement 2.** i.p. injection of TEPP-46 alleviated the development of Experimental Autoimmune Encephalomyelitis (EAE).

**Figure supplement 2—source data 1.** Source data for *Figure 7—figure supplement 2A, C and E*.

the reduced inflammation, demyelination, activation of astrocytes and microglia. Moreover, the in vivo effect of TRIM21 on promoting the nuclear translocation of PKM2 were verified, as evidenced by the observed inhibition of PKM2 nuclear immunostaining in astrocytes in shTRIM21-treated EAE mice. Our in vivo results suggested that targeting TRIM21-PKM2 is a promising approach for clinical

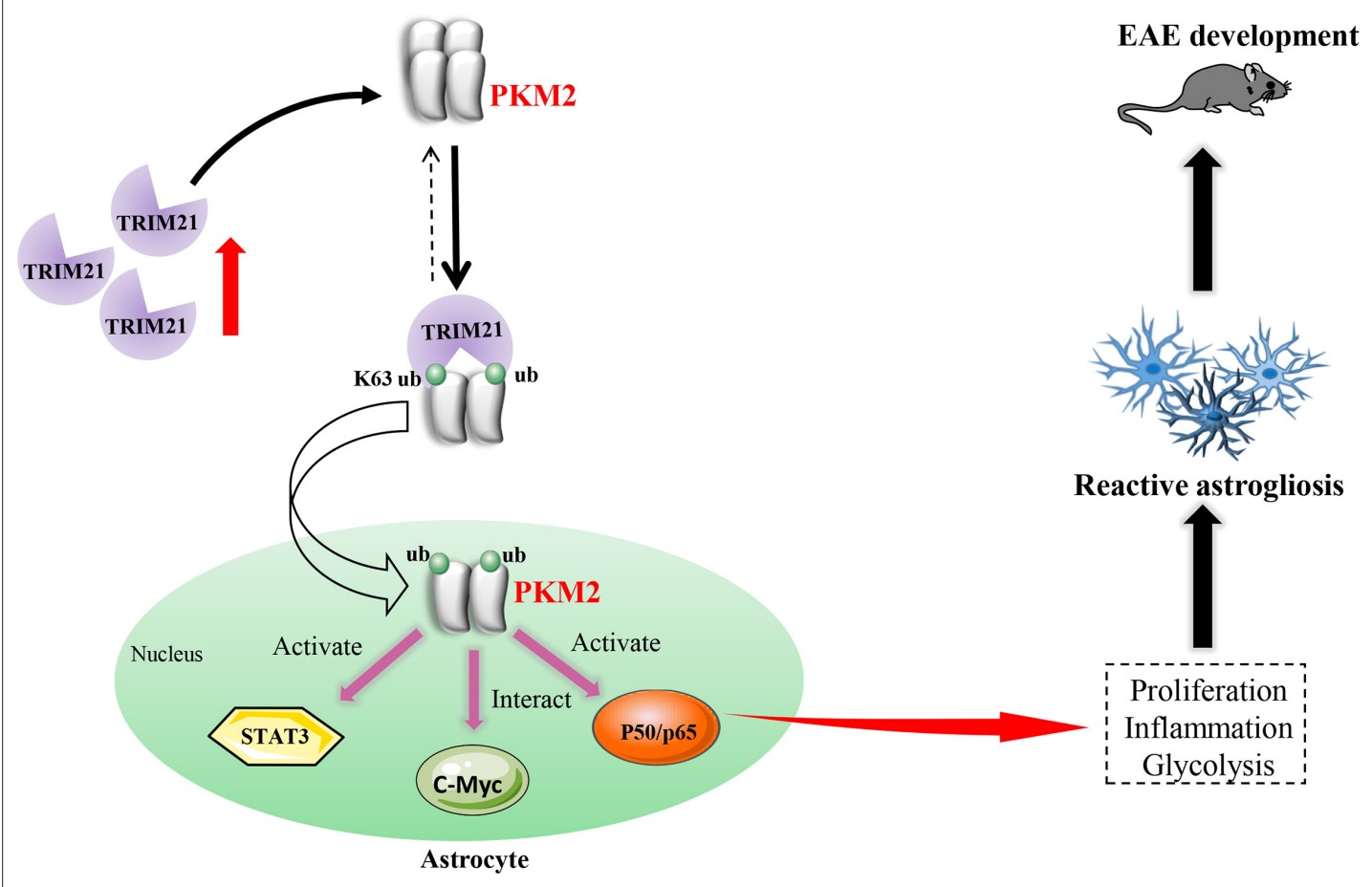

**Figure 8.** Schematic proposal of nuclear translocation of PKM2 in astrocytes of EAE. In astrocyte of EAE mice, TRIM21 expression is upregulated. E3 ubiquitin ligase TRIM21 ubiquitylates PKM2 and promotes its nuclear translocation, nuclear PKM2 activated STAT3 and NF-$\kappa$B pathways and interact with c-Myc to enhance glycolysis and proliferation in astrocytes. Thus, TRIM21-PKM2 pathway exerts a potential role in activating astrocytes and inducing EAE development.

treatment of multiple sclerosis. However, whether TRIM21-PKM2 could potentially serve as therapeutic targets in multiple sclerosis warrants further studies.

In conclusion, our study revealed that PKM2 nuclear translocation is the key mechanism accounting for glycolysis-dominant metabolic switch and proliferation of astrocytes. We propose a post-translational modification mechanism for the regulation of PKM2 nuclear translocation by the ubiquitin ligase TRIM21. From the perspective of metabolism, our study provides a rationale for targeting glycolysis metabolism to ameliorate astrocyte-mediated CNS diseases.

## Materials and methods
### Animal experiments

Eight-week-old female C57BL/6 mice were obtained and kept in the animal center of Shaanxi Normal university. All experimental procedures complied with Committee for Research and Animal Ethics of Shaanxi Normal university. The induction of EAE model was conducted as previously described (*Yan et al., 2012*). Briefly, mice were immunized on the back with 200 µg MOG 35–55 peptide (MOG$_{35-55}$) in 200 µL Complete Freund's adjuvant (CFA). CFA was prepared by resuspending 10 mg Mycobacterium tuberculosis H37Ra (Becton Dickinson, MD, USA) in 1 mL incomplete Freund's adjuvant (Sigma-Aldrich, St. Louis, MO, USA). A total of 200 ng pertussis toxin (List Biological Laboratories, CA, USA) dissolved in PBS was given intraperitoneally (i.p.) on day 0 and 2 post-immunization (p.i.). For TEPP-46 treatment, mice were injected intraperitoneally (i.p) with 200 µl vehicle (5% DMSO, 30% PEG300,

5% Tween 80 and 60% ddH$_2$O) or 50 mg/kg TEPP-46 dissolved in vehicle every other day from day 0 to day 8 p.i. All mice were divided into experimental groups randomly. All scoring processes were double-blinded. Mice were scored daily as follows: 0, no clinical symptoms; 1, paralyzed tail; 2, paralysis of one hind limb; 3, paralysis of two hind limbs; 4, paralysis of trunk; 5, death. Spinal cord tissue was collected at the onset (score 1; Day 7–17 p.i.), peak (score ≥3; Day 14–24 p.i.) and chronic stages of EAE (score ≥2; Day 21–26 p.i).

## In vivo injection of lentivirus

For in vivo injection of lentivirus, mice were anaesthetized and placed on a stereotaxic frame. A total of 1×10$^7$ IU/mouse shTRIM21 or control virus was injected with microsyringe at the following coordinates: 2.0 mm lateral, 1.0 mm caudal to bregma, and 2.5 mm below the skull surface. Twenty µl lentivirus was delivered at 1 µl/min. After each injection, the syringe was left for 10 min and then withdrawn slowly.

## Splenocyte isolation and supernatant of MOG$_{35-55}$-stimulated-splenocytes

Splenocytes were isolated from EAE mice 15 d (disease onset) after MOG$_{35-55}$ immunization. Briefly, spleen cells were suspended in RPMI-1640 medium containing 10% FBS. Splenocytes were plated in 12-well plates at 1×10$^6$ cells/well containing 50 µg/mL MOG$_{35-55}$ and cultured at 37 °C in 5% CO$_2$. After stimulation for 60 hr, cell suspension was centrifuged at 3000 rpm for 5 min and supernatants were collected. For the culture of MOG$_{sup}$-stimulated astrocytes, astrocytes were grown in medium containing 70% DMEM supplemented with 10% FBS and 30% supernatant from MOG$_{35-55}$-stimulated-splenocytes.

## Isolation and culture of primary astrocytes

Neonatal mice were killed and neuronal tissues were dissociated using Neural Tissue Dissociation Kit (Miltenyi Biotech, Auburn, CA) according to the manufacturer's instructions. Cell suspension was centrifuged at 800 × $g$ for 10 min. Subsequently, astrocytes were separated using anti-GLAST microbead kit (Miltenyi Biotech, Auburn, CA). Cells were suspended in buffer and 20 µL anti-Glast-Biotin was added. After incubation for 15 min at 4 °C, 3 mL buffer was added, followed by centrifuging at 800 × $g$ for 10 min. The supernatant was then aspirated, and 20 µL anti-Biotin was added, mixed well, and incubated at 4 °C for 15 min. The separation column was rinsed with 3 mL buffer three times, and the solution was centrifuged at 400 × $g$ for 10 min. Cells were suspended, counted and seeded in 60 mm dishes and grown in DMEM supplemented with 10% FBS. Purity of astrocytes was >95% as determined by GFAP immunostaining.

## Single-cell RNA sequencing

We prepared cells from mouse brain tissues by using adult brain dissociation kit (Miltenyi) according to the manufacturer's instruction. Briefly, cells with more than 90% viability were loaded onto the controller to generate single-cell gel bead emulsions. Single-cell RNA-seq libraries were prepared using version 3 Chromium Single Cell 3' Library (10×Genomics, available here). Sequencing were performed on Illumina NovaSeq6000. We used Cell Ranger version 4.0.0 to process raw sequencing data, barcode processing and single-cell UMI (unique molecular index) counting. Sequencing data have been deposited into the Gene Expression Omnibus (GEO) under the accession number GSE263883.

## Immunoprecipitation

Indicated antibodies (anti-PKM2, TRIM21, Flag, Myc and IgG) were incubated separately with Dynabeads M-270 Epoxy (Thermo Fisher Scientific) on a roller at 37 °C overnight to generate antibody-conjugated beads. Cell samples were lysed with ice-cold extraction buffer (Thermo Fisher Scientific) containing protease inhibitors. One-tenth of the supernatant was retained as input. Remaining supernatants were incubated with appropriate antibody-conjugated magnetic beads on a roller at 4 °C for 1 hr. Precipitates were washed and subjected to subsequent western blotting analysis.

## Mass spectrometry

Primary astrocytes were treated with supernatants from MOG$_{35-55}$-stimulated splenocytes for 24 hr before collection. Cells without any treatments and MOG$_{sup}$-stimulated primary astrocytes were lysed.

Cell lysates were immunoprecipitated with antibody against PKM2, and mass spectrometry was used to identify PKM2-interacting proteins with Q-Exactive Orbitrap mass spectrometer (Thermo Fisher) by Shanghai Bioprofile Biotechnology Co., Ltd. Peptide identification was performed by MaxQuant software, with a Uniprot mouse protein sequence database (55286 entries).

## Lentivirus-mediated short hairpin RNA interference and overexpression

Mir-30 based lentiviral vector with GFAP promoter was constructed as previously described. XhoI and EcoRI sites were used for cloning small hairpin RNAs (shRNAs; *Yan et al., 2012*).

Target sequences for sh*TRIM21* were as follows: shTRIM21-1:5′-GGAGCCTATGAGTATCGAATG-3′, shTRIM21-2:5′-GGAAAGAGTTGGCCGAGAAGA-3′, shTRIM21-3:5′-GCTCCCTCATTTACACCTTCT-3′, shControl: 5′- cctaaggttaagtcgccctcg-3′.

Primary astrocytes were cultured in six-well plates and infected with shTRIM21 or shControl. For overexpression of TRIM21, lentiviral vector with GFAP-promoter was used. TRIM21 cDNA was subcloned into lentivirus vector. Primary cultures were infected with LV-NC or LV-TRIM21.

## Glucose consumption and lactate production assays

The indicated cells ($1\times10^4$ per well) were seeded into 96-well plates and cultured for 24 hr. Cells were starved for 12 hr in serum-free DMEM medium supplemented with low glucose. With corresponding treatments, the supernatant was collected. Glucose consumption was determined using glucose oxidase method (Applygen Technologies, Beijing, China). The levels of lactate production were determined using lactate assay kit (Nanjing Jiancheng Bioengineering Institute, Nanjing, China). Glucose consumption and lactate production were normalized to cell numbers.

## Immunofluorescence

For tissue immunofluorescence staining, cryosections were blocked with buffer containing 1% BSA and 0.3% Triton X-100 at room temperature (RT) for 1 hr. Then, sections were incubated with primary antibodies anti-PKM2 (bs-0101R, Bioss), anti-GFAP (EM140707, Huabio), anti-TRIM21 (12108–1-AP, Proteintech), anti-MBP (10458–1-AP, Proteintech), or anti-IBA1 (019–19741, Wako) overnight at 4 °C. Then the Alexa Fluor 488 or Cy3-conjugated secondary antibodies (Zhuangzhibio, Xi'an, China) were applied at room temperature for 1 hr. Cell nuclei were labeled with DAPI.

For cell immunochemistry, cells cultured on glass coverslips were fixed with 4% PFA for 10 min at RT, followed by permeabilization with 0.3% Triton X-100. Non-specific binding was blocked with buffer containing 3% BSA for 30 min at RT. Briefly, samples were then incubated with primary antibodies and secondary antibodies. DAPI was used to stain the nuclei. Tissues and cells were observed and images were acquired using an EVOS FL Auto 2 Cell image system (Invitrogen). The fluorescence intensity was measured by ImageJ.

## Hematoxylin-Eosin (HE) and Luxol Fast Blue (LFB) staining

Spinal cords were dissected, fixed in 4% paraformaldehyde at 4°C, and 7-μm-thin paraffin sections were prepared on slides. After dewaxing, sections were dyed with hematoxylin for 2 min, and eosin for 2 min previous to water wash. Sections were then dehydrated through ethanol gradients (from 75% to 100%), cleared in xylene and mounted with neutral resin.

For LFB staining, cryosections of spinal cord were cut at a thickness of 7 μm. Briefly, slices were stained in LFB staining solution (Servicebio) for 12 hr at 56 °C. After rinsing in 95% ethanol, sections were decolorized in 0.05% lithium carbonate for 20 sr and rinsed in 70% ethanol for 20 sr. Sections were then dehydrated and mounted as described for HE staining.

## Cell proliferation assays

Cells were plated at a density of $5\times10^4$ per well and cultured overnight. After treatment, the proliferation of astrocytes was assessed by an EdU-488 or EdU-594 cell proliferation detection kit (Beyotime, C0071S and C0078S). For Cell Counting Kit-8 analysis, cells were seeded separately in each 96-well plate and cultured for 24 hr, 48 hr, and 72 hr, respectively. One hour before the endpoint of incubation, 10 μl CCK-8 reagent was added, $OD_{450nm}$ value was determined by Infinite F50 (Tecan) microplate reader.

## Protein extraction and western blotting

Cells were lysed in RIPA buffer supplemented with proteinase inhibitor cocktail (Topscience, Shanghai, China). Whole cell lysates were obtained after centrifugation. Nuclear protein was extracted using Nuclear and cytoplasmic Extraction Kit (Solarbio, Beijing, China) according to the manufacturer's instructions. Protein concentrations were determined by using a BCA kit, and then subjected to western blotting. Protein samples were separated by SDS-PAGE and transferred onto PVDF membranes. After being blocked for 2 hr in 5% skim-milk buffer, membranes were incubated overnight at 4 °C with the following primary antibodies: antibodies against PKM2 (1:5000, 60268–1-Ig, Proteintech), phospho-c-myc (1:500, ET1609-64, Huabio), c-myc (1:1000, CPA1778, Cohesion Biosciences), LDHA (1:1000, ET1608-57, Huabio), STAT3 (1:5000, 60199–1-Ig, Proteintech), phospho-STAT3 (1:1000, bs-1658R, Bioss), phospho-p65 (1:1000, GB113882-100, Servicebio), p65 (1:1000, CPA2000, Cohesion Biosciences), phospho-IKK(1:1000, bs-3237R, Bioss), IKK (1:1000, GB11292-1-100, ServiceBio), Lamin (1:1000 CPA1693, Cohesion Biosciences), Tubulin (11224–1-AP, Proteintech), Flag (1:1000, AE004, Abclonal), Myc (1:3000, AE010, Abclonal), TRIM21 (1:1000, 12108–1-AP, Proteintech), and β-actin (1:2000, GB12001-100, ServiceBio). Membranes were then washed and probed with HRP conjugated secondary antibodies. Membranes were visualized with ECL detection system (Tanon 4600, Shanghai, China).

## RNA extraction and qPCR

Total RNA was extracted with TRIzol and cDNA was synthesized by reverse transcription (DEEYEE, Shanghai, China). qPCR was performed by using 2×qPCR SmArt Mix (DEEYEE, Shanghai, China) with StepOnePlus Real-time PCR system (Thermo Fisher). The fold-change data were obtained using the delta-delta Ct method (*Livak and Schmittgen, 2001*). Primers used in this study were listed in *Supplementary file 1*.

## Statistics

Data were analyzed with GraphPad Prism software (version 8.0). Differences between two groups were analyzed using two-tailed Student's t-test. Differences between more than two groups were determined by one-way ANOVA with Dunnett's post-hoc test. Mean clinical scores of animals were determined by two-way ANOVA analysis. A p value of<0.05 indicated significant differences between groups.

# Acknowledgements

We thank Shanghai Bioprofile Biotechnology Co., Ltd for mass spectrometry and bioinformatics analysis. This work was supported by National Natural Science Foundation of China (No. 82071348), Natural Science Basic Research Program of Shaanxi (No. 2023-JC-JQ-64), Fundamental Research Funds for the Central University (No. GK202304034).

# Additional information

## Funding

| Funder | Grant reference number | Author |
| --- | --- | --- |
| National Natural Science Foundation of China | 82071348 | Yaping Yan |
| Natural Science Basic Research Program of Shaanxi Province | 2023-JC-JQ-64 | Yaping Yan |
| Fundamental Research Funds for the Central Universities | GK202304034 | Yang Yang |

The funders had no role in study design, data collection and interpretation, or the decision to submit the work for publication.

## Author contributions
Luting Yang, Conceptualization, Data curation, Formal analysis, Validation, Methodology, Writing – original draft, Project administration; Chunqing Hu, Investigation, Writing - review and editing; Xiaowen Chen, Yanxin Xiao, Weitai He, Investigation; Jie Zhang, Validation, Investigation; Zhe Feng, Software; Tingting Cui, Xin Zhang, Yaling Zhang, Methodology; Yang Yang, Funding acquisition, Methodology; Yaping Yan, Supervision, Funding acquisition, Methodology

## Author ORCIDs
Luting Yang ⓘ https://orcid.org/0000-0002-7180-8328
Chunqing Hu ⓘ https://orcid.org/0009-0007-0839-3346
Yaping Yan ⓘ https://orcid.org/0000-0002-1714-2318

## Ethics
The study was conducted in accordance with the Declaration of Helsinki, all experimental procedures complied with Committee for Research and Animal Ethics of Shaanxi Normal university (Permit Number: 2023-055).

Reviewer #1 (Public Review): https://doi.org/10.7554/eLife.98181.3.sa1
Reviewer #2 (Public Review): https://doi.org/10.7554/eLife.98181.3.sa2
Reviewer #3 (Public Review): https://doi.org/10.7554/eLife.98181.3.sa3
Reviewer #4 (Public Review): https://doi.org/10.7554/eLife.98181.3.sa4
Author response https://doi.org/10.7554/eLife.98181.3.sa5

# Additional files

## Supplementary files
- MDAR checklist
- Supplementary file 1. List of primers used in this study.

## Data availability
Sequencing data have been deposited into the Gene Expression Omnibus (GEO) under the accession number GSE263883. All data generated or analysed during this study are included in the manuscript and supporting files. Source data files have been provides for *Figure 1* to *Figure 7*, and for *Figure 1—figure supplement 1*, *Figure 2—figure supplements 1 and 2*, *Figure 3—figure supplement 1*, *Figure 6—figure supplement 1*, *Figure 7—figure supplement 2*.

The following dataset was generated:

| Author(s) | Year | Dataset title | Dataset URL | Database and Identifier |
| --- | --- | --- | --- | --- |
| Tingting C, Yang L, Hu C, Chen X, Zhang J, Feng Z, Xiao Y, He W, Cui T, Zhang X, Yang Y, Zhang Y | 2024 | Upregulated expression of ubiquitin ligase TRIM21 promotes PKM2 nuclear translocation and astrocyte activation in experimental autoimmune encephalomyelitis | https://www.ncbi.nlm.nih.gov/geo/query/acc.cgi?acc=GSE263883 | NCBI Gene Expression Omnibus, GSE263883 |

The following previously published dataset was used:

| Author(s) | Year | Dataset title | Dataset URL | Database and Identifier |
| --- | --- | --- | --- | --- |
| Boddeke EW, Kooistra SM, Eggen BJ | 2021 | Transcriptional profiling of astrocyte subtypes in naive mice and during experimental autoimmune encephalomyelitis | https://www.ncbi.nlm.nih.gov/geo/query/acc.cgi?acc=GSE136358 | NCBI Gene Expression Omnibus, GSE136358 |

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
