## [Editor Report · eLife assessment]

This **important** work describes the activation of astrocytes via the nuclear translocation of PKM2 in an animal model of multiple sclerosis. This study provides **convincing** evidence of the interaction between TRIM21 and PKM2 as the crucial molecular event leading to the translocation of PKM2 and a metabolic shift towards glycolysis dominance, fostering proliferation in stimulated astrocytes. This finding is significant as it underscores the potential of targeting glycolytic metabolism to mitigate neurological diseases mediated by astrocytes, offering a strong rationale for potential therapeutic interventions.

---

## [Referee Report · Reviewer #1 (Public Review)]

Summary:

Yang. Hu et al. investigated the molecular mechanism that cause astrocyte activation and its implications for multiple sclerosis. This study focuses on the enzyme PKM2, known for its role in glycolysis, and its nuclear translocation in reactive astrocytes in a mouse model of multiple sclerosis (EAE). Preventing the nuclear translocation of PKM2 reduces astrocyte activation, proliferation, glycolysis, and inflammatory cytokine secretion. Importantly, the study reveals that TRIM21 controls PKM2's nuclear translocation through ubiquitination, promoting its nuclear import and enhancing its activity. Single-cell RNA sequencing and immunofluorescence confirm TRIM21 upregulation in EAE astrocytes, and alteration of TRIM21 levels affect PKM2-dependent glycolysis and proliferation. Their findings suggest that targeting the TRIM21-PKM2 axis could be a therapeutic strategy for treating neurological diseases involving astrocyte activation.

Strength:

This work provides a comprehensive exploration of PKM2's nuclear role and its interaction with TRIM21 in EAE, offering new insights for therapeutic strategies targeting metabolic reprogramming in astrocyte activation. The strength of the study is the use of advanced techniques such as single cell RNA sequencing, in vitro and in vivo knockdown techniques to support the data. With the addition of new data and explanations in the manuscript, the authors have rendered their claimed ideas more supportive.

Weakness:

The revisions and implementation of suggestions have greatly improved the overall quality of the manuscript. I would like to thank the authors for carefully evaluating all the suggestions and for providing extra explanations and response figures. However, there are still some points that need to be corrected and clarified.

---

## [Referee Report · Reviewer #2 (Public Review)]

This study significantly advances our understanding of the metabolic reprogramming underlying astrocyte activation in neurological diseases such as multiple sclerosis. By employing an experimental autoimmune encephalomyelitis (EAE) mouse model, the authors discovered a notable nuclear translocation of PKM2, a key enzyme in glycolysis, within astrocytes. Preventing this nuclear import via DASA 58 substantially attenuated primary astrocyte activation, characterized by reduced proliferation, glycolysis, and inflammatory cytokine secretion.

Moreover, the authors uncovered a novel regulatory mechanism involving the ubiquitin ligase TRIM21, which mediates PKM2 nuclear import. TRIM21 interaction with PKM2 facilitated its nuclear translocation, enhancing its activity in phosphorylating STAT3, NFκB, and c-myc. Single-cell RNA sequencing and immunofluorescence staining further supported the upregulation of TRIM21 expression in astrocytes during EAE.

Manipulating this pathway, either through TRIM21 overexpression in primary astrocytes or knockdown of TRIM21 in vivo, had profound effects on disease severity, CNS inflammation, and demyelination in EAE mice. This comprehensive study provides invaluable insights into the pathological role of nuclear PKM2 and the ubiquitination-mediated regulatory mechanism driving astrocyte activation.

The author's use of diverse techniques, including single-cell RNA sequencing, immunofluorescence staining, and lentiviral vector knockdown, underscores the robustness of their findings and interpretations. Ultimately, targeting this PKM2-TRIM21 axis emerges as a promising therapeutic strategy for neurological diseases involving astrocyte dysfunction.

While the strengths of this piece of work are undeniable, some concerns could be addressed to refine its impact and clarity further.

---

## [Referee Report · Reviewer #3 (Public Review)]

Summary:

Pyruvate kinase M2 (PKM2) is a rate limiting enzyme in glycolysis and its translocation to nucleus in astrocytes in various nervous system pathologies has been associated with a metabolic switch to glycolysis which is a sign of reactive astrogliosis. Authors investigated whether this occurs in experimental autoimmune encephalomyelitis (EAA), an animal model of multiple sclerosis (MS). They show that in EAA, PKM2 is ubiquitinated by TRIM21 and transferred to the nucleus in astrocytes. Inhibition of TRIM21-PKM2 axis efficiently blocks reactive gliosis and partially alleviates symptoms of EAA. Authors conclude that this axis can be a potential new therapeutic target in the treatment of MS.

Strengths:

The study is well-designed, controls are appropriate and a comprehensive battery of experiments has been successfully performed. Results of in vitro assays, single cell RNA sequencing, immunoprecipitation, RNA interference, molecular docking and in vivo modeling etc. complement and support each other.

Weaknesses:

Though EAA is a valid model of MS, a proposed new therapeutic strategy based on this study needs to have support from human studies.

The comments above are still valid for this revised version of the manuscript.

---

## [Referee Report · Reviewer #4 (Public Review)]

The authors report the role of the Piruvate Kinase M2 (PKM2) enzyme nuclear translocation as fundamental in the activation of astrocytes in a model of autoimmune encephalitis (EAE). They show that astrocytes, activated through culturing in EAE splenocytes medium, increase their nuclear PKM2 with a consequent activation of NFkB and STAT3 pathways. Prevention of PKM2 nuclear translocation decreases astrocyte counteracts this activation. The authors found that the E3 ubiquitin ligase TRIM21 interacts with PKM2 and promotes its nuclear translocation. In vivo, either silencing of TRIM21 or inhibition of PKM2 nuclear translocation ameliorates the severity of the disease in the EAE model.

Strengths

This work contributes to the knowledge of the complex action of the PKM2 enzyme in the context of an autoimmune-neurological disease, highlighting its nuclear role and a novel partner, TRIM21, promoting its nuclear translocation. In vivo amelioration of the pathological signs through inhibition of either of the two, PKM2 and TRIM21, provides a novel rationale for therapeutic targeting.

Weaknesses

I believe that the major weakness is the fact that TRIM21 is known to have per se many roles in autoimmune and immune pathways and some of the effects observed might be due to a PKM2-independent action. Some of the experiments to link the two proteins, besides their interaction, are not completely clarifying the issue. On top of that, the in vivo experiments address the role of TRIM21 and the nuclear localisation of PKM2 independently, thus leaving the matter unsolved.

In general, the conclusions of the manuscript are supported by the reported results. The points to be addressed in future are the assessment of PKM2 as substrate of TRIM21 ubiquitin ligase activity and the proof of the epistatic relationship of TRIM21 and PKM2 in astrocyte activation. However, the data surely open novel directions to follow for the understanding of multiple sclerosis and related pathologies.

---

## [Author Response]

The following is the authors’ response to the original reviews.

**Public Reviews:**

**Reviewer #1 (Public Review):**
Summary:Yang, Hu et al. examined the molecular mechanisms underlying astrocyte activation and its implications for multiple sclerosis. This study shows that the glycolytic enzyme PKM2 relocates to astrocyte nuclei upon activation in EAE mice. Inhibiting PKM2's nuclear import reduces astrocyte activation, as evidenced by decreased proliferation, glycolysis, and inflammatory cytokine release. Crucially, the study identifies TRIM21 as pivotal in regulating PKM2 nuclear import via ubiquitination. TRIM21 interacts with PKM2, promoting its nuclear translocation and enhancing its activity, affecting multiple signaling pathways. Confirmatory analyses using single-cell RNA sequencing and immunofluorescence demonstrate TRIM21 upregulation in EAE astrocytes. Modulating TRIM21 expression in primary astrocytes impacts PKM2-dependent glycolysis and proliferation. In vivo experiments targeting this mechanism effectively mitigate disease severity, CNS inflammation, and demyelination in EAE.The authors supported their claims with various experimental approaches, however, some results should be supported with higher-quality images clearly depicting the conclusions and additional quantitative analyses of Western blots.

Thanks for the reviewer’s comments. We agree with the reviewer and have added higher magnification images, for example Fig.2A to better visualize the localization of PKM2 in DASA-treated conditions, and Fig. 3A and Fig.3B to better visualize the pSTAT3 and pp65. Moreover, we have added quantitative analyses of Western blots for some key experiments, for example quantitative results for Fig.2D is added in Fig.S3 to show the change of PKM2 and p-c-myc in DASA-58-treated conditions and quantitative results for Fig. 3D are added in Fig.S4B and S4C to show the change of nuclear and cytoplasmic PKM2, STAT3 and NF-κB in different conditions.

Strength:This study presents a comprehensive investigation into the function and molecular mechanism of metabolic reprogramming in the activation of astrocytes, a critical aspect of various neurological diseases, especially multiple sclerosis. The study uses the EAE mouse model, which closely resembles MS. This makes the results relevant and potentially translational. The research clarifies how TRIM21 regulates the nuclear import of PKM2 through ubiquitination by integrating advanced techniques. Targeting this axis may have therapeutic benefits since lentiviral vector-mediated knockdown of TRIM21 in vivo significantly reduces disease severity, CNS inflammation, and demyelination in EAE animals.

We thank the reviewer for their positive and constructive comments on the manuscript.

Weaknesses:The authors reported that PKM2 levels are elevated in the nucleus of astrocytes at different EAE phases compared to cytoplasmic localization. However, Figure 1 also shows elevated cytoplasmic expression of PKM2. The authors should clarify the nuclear localization of PKM2 by providing zoomed-in images. An explanation for the increased cytoplasmic PKM2 expression should provided. Similarly, while PKM2 translocation is inhibited by DASA-58, in addition to its nuclear localization, a decrease in the cytoplasmic localization of PKM2 is also observed. This situation brings to mind the possibility of a degradation mechanism being involved when its nuclear translocation of PKM2 is inhibited.

According to the results of immunofluorescence staining of PKM2 in spinal cord of EAE mice and in cultured primary astrocytes, in addition to the observation of PKM2 nuclear translocation in EAE conditions, we showed an elevated expression of PKM2 in astrocytes, including the cytoplasmic and nuclear expression. In neurological diseases, various studies showed consistent results, for example, following spinal cord injury (SCI), not only the upregulated expressing of PKM2 but also nuclear translocation was observed in astrocytes (Zhang et al., 2015). In EAE conditions, CNS inflammation is elevated and several proinflammatory cytokines and chemokines might contribute to the upregulated expression of PKM2 in astrocytes. We have tested TNFα and IL-1β, which are recognized to play important roles in EAE and MS (Lin and Edelson, 2017, Wheeler et al., 2020), and results from western blots showed the increased expression of PKM2 upon stimulation with TNFα and IL-1β (Author response image 1). Moreover, according to the reviewer’s suggestions, we have added zoomed-in images for figure 2A.

Additionally, the reviewer has noted the decrease in the cytoplasmic PKM2 level, degradation-related mechanism and other mechanisms might be involved in this process.

**Author response image 1. sa5fig1:** Upregulated expression of PKM2 in astrocytes following stimulation with TNF-α and IL-1β. Primary astrocytes were stimulated with TNF-α and IL-1β (50 ng/mL) for 48 h and western blotting analysis were performed.

In Figure 3D, the authors claim that PKM2 expression causes nuclear retention of STAT3, p65, and p50, and inhibiting PKM2 localization with DASA-58 suppresses this retention. The western blot results for the MOG-stimulated group show high levels of STAT3, p50, and p65 in nuclear localization. However, in the MOG and DASA-58 treated group, one would expect high levels of p50, p65, and STAT3 proteins in the cytoplasm, while their levels decrease in the nucleus. These western blot results could be expanded. Additionally, intensity quantification for these results would be beneficial to see the statistical difference in their expressions, especially to observe the nuclear localization of PKM2.

We agree with the reviewer’s comments and we have incorporated the quantification of STAT3，p50 and p65 for Fig.3D and Fig.S4B and Fig.S4C. Nevertheless, given that DASA-58 did not trigger a notable increase in the cytoplasmic level of PKM2, we did not detect an upregulation of STAT3, p50, or p65 in the cytoplasm of the MOG and DASA-58-treated groups. With the quantification results, it is more obvious to see the changes of these proteins in different conditions.

The discrepancy between Figure 7A and its explaining text is confusing. The expectation from the knocking down of TRIM21 is the amelioration of activated astrocytes, leading to a decrease in inflammation and the disease state. The presented results support these expectations, while the images showing demyelination in EAE animals are not highly supportive. Clearly labeling demyelinated areas would enhance readers' understanding of the important impact of TRIM21 knockdown on reducing the disease severity.

Thank you for pointing this out. We sincerely apologize for our carelessness. Based on your comments, we have made the corrections in the manuscript. As there is indeed a statistical difference in the mean clinical scores between shTRIM21-treated group and shVec group, we have accordingly revised the sentence for Figure 7A to state, “At the end time point at day 22 p.i., shTRIM21-treated group showed reduced disease scores compared to control groups (Fig. 7A).” .

Additionally, we have added the whole image of the spinal cord for MBP in Author Response image 2. Moreover, we have labelled the demyelinated areas to facilitate readers’ understanding.

**Author response image 2. sa5fig2:** MBP staining of the whole spinal cord in EAE mice from shVec and shTRIM21 group. Scale bar: 100 μm. Demyelinated areas are marked with dashed lines.

**Reviewer #2 (Public Review):**
This study significantly advances our understanding of the metabolic reprogramming underlying astrocyte activation in neurological diseases such as multiple sclerosis. By employing an experimental autoimmune encephalomyelitis (EAE) mouse model, the authors discovered a notable nuclear translocation of PKM2, a key enzyme in glycolysis, within astrocytes.Preventing this nuclear import via DASA 58 substantially attenuated primary astrocyte activation, characterized by reduced proliferation, glycolysis, and inflammatory cytokine secretion.Moreover, the authors uncovered a novel regulatory mechanism involving the ubiquitin ligase TRIM21, which mediates PKM2 nuclear import. TRIM21 interaction with PKM2 facilitated its nuclear translocation, enhancing its activity in phosphorylating STAT3, NFκB, and c-myc. Single-cell RNA sequencing and immunofluorescence staining further supported the upregulation of TRIM21 expression in astrocytes during EAE.Manipulating this pathway, either through TRIM21 overexpression in primary astrocytes or knockdown of TRIM21 in vivo, had profound effects on disease severity, CNS inflammation, and demyelination in EAE mice. This comprehensive study provides invaluable insights into the pathological role of nuclear PKM2 and the ubiquitination-mediated regulatory mechanism driving astrocyte activation.The author's use of diverse techniques, including single-cell RNA sequencing, immunofluorescence staining, and lentiviral vector knockdown, underscores the robustness of their findings and interpretations. Ultimately, targeting this PKM2-TRIM21 axis emerges as a promising therapeutic strategy for neurological diseases involving astrocyte dysfunction.While the strengths of this piece of work are undeniable, some concerns could be addressed to refine its impact and clarity further; as outlined in the recommendations for the authors.

Thanks for the reviewer’s comment and positive evaluation of our present work. We have further answered each question in recommendations section.

**Reviewer #3 (Public Review):**
Summary:Pyruvate kinase M2 (PKM2) is a rate-limiting enzyme in glycolysis and its translocation to the nucleus in astrocytes in various nervous system pathologies has been associated with a metabolic switch to glycolysis which is a sign of reactive astrogliosis. The authors investigated whether this occurs in experimental autoimmune encephalomyelitis (EAA), an animal model of multiple sclerosis (MS). They show that in EAA, PKM2 is ubiquitinated by TRIM21 and transferred to the nucleus in astrocytes. Inhibition of TRIM21-PKM2 axis efficiently blocks reactive gliosis and partially alleviates symptoms of EAA. Authors conclude that this axis can be a potential new therapeutic target in the treatment of MS.Strengths:The study is well-designed, controls are appropriate and a comprehensive battery of experiments has been successfully performed. Results of in vitro assays, single-cell RNA sequencing, immunoprecipitation, RNA interference, molecular docking, and in vivo modeling etc. complement and support each other.Weaknesses:Though EAA is a valid model of MS, a proposed new therapeutic strategy based on this study needs to have support from human studies.

We agree that although we have clarified the therapeutic potential of targeting TRIM21 or PKM2 in the treatment of EAE, a mouse model of MS, the application in human studies warrants further studies. While considering the use of TRIM21 as a target for treating multiple sclerosis in clinical trials, several issues need to be addressed to ensure the safety, efficacy and feasibility. One such aspect is the development of drug that specifically target TRIM21 in brain, capable of crossing the blood-brain barrier and have minimal off-target effects. The translation of preclinical finding into clinical trials poses a significant challenge. To provide evidence for the similarities between the EAE model and multiple sclerosis, we have screened GEO databases (Author response image 3). In GSE214334 which analyzed transcriptional profiles of normal-appearing white matter from non-MS and different subtypes of disease (RRMS, SPMS and PPMS). Although no statistical difference was observed among different groups, the TRIM21 expression has tendency to increase in SPMS (secondary progressive MS) and PPMS (primary progressive MS) patients. In GSE83670, astrocytes from 3 control white matter and 4 multiple sclerosis normal appearing white matter (NAWM) were analyzed. TRIM21 mRNA expression is higher in MS group (78.73 ± 10.44) compared to control group (46.67 ± 24.15). Although these two GEO databases did not yield statistically significant differences, TRIM21 expression appears to be elevated in the white matter of MS patients compared to controls.

To address this limitation, we have incorporated the following statement in the discussion section: “However, whether TRIM21-PKM2 could potentially serve as therapeutic targets in multiple sclerosis warrants further studies.”

**Author response image 3. sa5fig3:** TRIM21 expression in control and MS patients based on published GEO database. (A) The expression of TRIM21 in normal-appearing white matter in non-MS (Ctl) and different clinical subtypes of MS (RRMS, SPMS, PPMS) based on GSE214334 (one-way ANOVA). (B) The expression of TRIM21 from multiple sclerosis normal appearing white matter (NAWM) and control WM based on GSE83670. RRMS, relapsing--remitting MS; SPMS, secondary progressive MS; PPMS, primary progressive MS (unpaired Student's t test). Data are represented as the means ± SEM.

**Reviewer #4 (Public Review):**
Summary:The authors report the role of the Pyruvate Kinase M2 (PKM2) enzyme nuclear translocation as fundamental in the activation of astrocytes in a model of autoimmune encephalitis (EAE). They show that astrocytes, activated through culturing in EAE splenocytes medium, increase their nuclear PKM2 with consequent activation of NFkB and STAT3 pathways. Prevention of PKM2 nuclear translocation decreases astrocyte counteracts this activation. The authors found that the E3 ubiquitin ligase TRIM21 interacts with PKM2 and promotes its nuclear translocation. In vivo, either silencing of TRIM21 or inhibition of PKM2 nuclear translocation ameliorates the severity of the disease in the EAE model.Strengths:This work contributes to the knowledge of the complex action of the PKM2 enzyme in the context of an autoimmune-neurological disease, highlighting its nuclear role and a novel partner, TRIM21, and thus adding a novel rationale for therapeutic targeting.Weaknesses:Despite the relevance of the work and its goals, some of the conclusions drawn would require more thorough proof:I believe that the major weakness is the fact that TRIM21 is known to have per se many roles in autoimmune and immune pathways and some of the effects observed might be due to a PKM2-independent action. Some of the experiments to link the two proteins, besides their interaction, do not completely clarify the issue. On top of that, the in vivo experiments address the role of TRIM21 and the nuclear localisation of PKM2 independently, thus leaving the matter unsolved.

We agree that TRIM21 has multifunctional roles and only some of their effects are due to PKM2-independent action. It is obvious that TRIM21 functions as ubiquitin ligases and its substrate are various. Here we identify PKM2 as one of its interacting proteins and our focus is the relationship between TRIM21 and the nuclear translocation PKM2, we have used diverse experiments to clarify their relationships, for example immunoprecipitation, western blotting, immunofluorescence, cyto-nuclear protein extraction. These aforementioned experiments are key points of our studies. From the results of in vitro experiments, targeting either TRIM21 or PKM2 might be potential targets for EAE treatment. Expectedly, from in vivo experiments, either targeting TRIM21 or PKM2 nuclear transport ameliorated EAE. In order to test the relationship of TRIM21 and PKM2 nuclear transport in vivo, we have stained PKM2 in shVec and shTRIM21-treated mice. Expectedly, knocking down TRIM21 led to a decrease in the nuclear staining of PKM2 in spinal cord astrocytes in EAE models (Figure S7A). This observation underscores that the therapeutic potential of inhibiting TRIM21 in astrocytes in vivo might be partially due to its role in triggering the reduced nuclear translocation of PKM2.

Some experimental settings are not described to a level that is necessary to fully understand the data, especially for a non-expert audience: e.g. the EAE model and MOG treatment; action and reference of the different nuclear import inhibitors; use of splenocyte culture medium and the possible effect of non-EAE splenocytes.

According to the reviewer’s suggestions, we have added more detailed descriptions in the materials and methods section, for example, the use of splenocytes culture medium, mass spectrometry, HE and LFB staining have been added. More details are incorporated in the part for “EAE induction and isolation and culture of primary astrocytes”. Moreover, the reference of DASA-58 in vitro and TEPP-46 in vivo as inhibitors of PKM2 nuclear transport were added.

The statement that PKM2 is a substrate of TRIM21 ubiquitin ligase activity is an overinterpretation. There is no evidence that this interaction results in ubiquitin modification of PKM2; the ubiquitination experiment is minimal and is not performed in conditions that would allow us to see ubiquitination of PKM2 (e.g. denaturing conditions, reciprocal pull-down, catalytically inactive TRIM21, etc.).

To prevent the misunderstanding, we have revised certain statements in the manuscript. In the updated version, the description is as follows: Hereby, we recognized PKM2 as an interacting protein of TRIM21, and further studies are required to determine if it is a substrate of E3 ligase TRIM21.

**Recommendations for the authors:**

**Reviewer #1 (Recommendations For The Authors):**
General recommendations:- The whole manuscript needs language editing.

We appreciate the comments of the reviewers. We have improved the writing of the manuscript. All modifications are underlined.

- Details of many experiments are not given in the materials and methods.

According to the reviewer’s suggestions, we have added more details for experiments in the materials and methods. For example, “Splenocyte isolation and supernatant of MOG35-55-stimulated-splenocytes”, “mass spectrometry”, “Hematoxylin-Eosin (HE) and Luxol Fast Blue (LFB) staining” were added in the section of Materials and Methods. More detailed information is given for EAE induction and isolation and culture of primary astrocytes.

- Line properties in graphics should be corrected, some lines in box plots and error bars are very weak and hardly visible. Statistical tests should be included in figure legends as well. Statistical differences should be mentioned for control vs DASA-58 (alone) in all related figures.

We have revised the figures to enhance their visibility by thickening the lines and error bars. In accordance with the reviewer’s suggestions, we have incorporated statistical tests in figure legends. Moreover, statistical analysis has been made among all groups, if there is no asterisk indicated in the figure legend and figure panels, it means there is no statistical difference between the control vs DASA-58 groups. For most of the experiments conducted in our studies, including lactate production, glucose consumption, the EdU analysis and CCK8 analysis, the change of STAT3 and NF-κB pathways, no statistical difference was observed between the control and DASA-58 group. The reason might be due to that in unstimulated astrocytes, the expression of PKM2 is low and nuclear translocation of PKM2 are few, which may explain why DASA-58 did not exert the anticipated effect. Thus, in our experiments, we have used MOGsup to stimulate astrocytes, enabling us to observe the impact of DASA-58 on the astrocyte proliferation and glycolysis in this condition.

- Scale bars, arrows, and labeling in the images are not visible.

We have improved the images according to the reviewer’s suggestions. The scale bars, arrows are made thicker and labeling are larger. The updated figures are visible.

- Quantitative analysis of all western blot results and their statistics could be provided in every image and for every protein.

For western blotting results which are further processed with quantitative analysis, for example, Fig.2D, fig. 5G, Fig. 6A and 6B, Fig. S4, we have added their statistics in the raw data sections. The other western blot results, for example, IP analysis, which are used to analyze protein-protein binding are not further processed with quantitative analysis.

- Proteins that are used for normalizations in western blots should be stated in the text.

We have added description of proteins that are used for normalization in western blots in figure legends. Moreover, in figure panels, proteins used for normalization are indicated. Globally, whole protein level is normalized to protein level of β-actin. For nuclear and cytoplasmic proteins, nuclear protein is normalized to the expression of lamin, cytoplasmic protein is normalized to the expression of tubulin.

- The manuscript investigates the role of TRIM21 in the nuclear localization of PKM2 in astrocytes in EAE mice, however almost no information is given about TRIM21 in the introduction. Extra information is given for PKM2, yet can be concisely explained.

We have added a paragraph that describes the information of TRIM21 in the introduction section. The description is as follows: “TRIM21 belongs to the TRIM protein family which possess the E3 ubiquitin ligase activity. In addition to its well-recognized function in antiviral responses, emerging evidences have documented the multifaceted role of TRIM21 in cell cycle regulation, inflammation and metabolism (Chen et al., 2022). Nevertheless, the precise mechanisms underlying the involvement of TRIM21 in CNS diseases remain largely unexplored.”

- "As such, deciphering glycolysis-dominant metabolic switch in astrocytes is the basis for understanding astrogliosis and the development of neurological diseases such as multiple sclerosis." The sentence could be supported by references.

To support this sentence, we have added the following references:

(1) Xiong XY, Tang Y, Yang QW. Metabolic changes favor the activity and heterogeneity of reactive astrocytes. Trends in endocrinology and metabolism: TEM 2022;33(6):390-400.

(2) das Neves SP, Sousa JC, Magalhães R, Gao F, Coppola G, Mériaux S, et al. Astrocytes Undergo Metabolic Reprogramming in the Multiple Sclerosis Animal Model. Cells 2023;12(20):2484.

Figure 1/Result 1:- Figure 1A-B: Quality of the images should be improved.

According to the reviewer’s suggestion, we have improved the quality of the image, images with higher resolution were added in figure 1A and figure 1B.

- Control images of Figure 1B are not satisfying. GFAP staining is very dim. Images from control cells should be renewed.

As mentioned by the reviewer’s, we have renewed the control images and added the DAPI staining figures for all groups. Compared with MOGsup stimulated astrocytes, the control cells are not in activated state and GFAP are relatively low.

- Labelings on the images are not sufficient, arrows and scale bars are not visible.

We have improved the images including labels, arrows and scale bars in all figures.

- How splenocytes were obtained from MOG induced mice were not given in the material and methods section. Thus, it should be clearly stated how splenocyte supernatant is generated (treatment details).

We have added the detailed information relating to splenocyte isolation and splenocyte supernatant entitled “Splenocyte isolation and supernatant of MOG35-55-stimulated-splenocytes” in the section of Materials and methods. “Splenocytes were isolated from EAE mice 15 d (disease onset) after MOG35-55 immunization. Briefly, spleen cells were suspended in RPMI-1640 medium containing 10% FBS. Splenocytes were plated in 12-well plates at 1x106 cells/well containing 50 μg/mL MOG35-55 and cultured at 37°C in 5% CO2. After stimulation for 60 h, cell suspension was centrifuged at 3000 rpm for 5 min and supernatants were collected. For the culture of MOGsup-stimulated astrocytes, astrocytes were grown in medium containing 70% DMEM supplemented with 10% FBS and 30% supernatant from MOG35-55-stimulated-splenocytes.”

- For general astrocyte morphology: authors showed the cells are GFAP+ astrocytes. It is surprising that these cells do not bear classical astrocyte morphology in cell culture. How long do you culture astrocytes before treatment? How do you explain their morphological difference?

Astrocytes were cultured for 2 to 3 weeks which correspond to 2-3 passages before treatment. There are several possible reasons for the morphological differences observed between GFAP+ astrocytes and their classical morphology. Firstly, the cell density. In low-density culture just as shown in Figure 1B, we have observed that astrocytes adopt a more flattened morphology. In high-density cultures, they adopt a stellate shape. Moreover, variations in culture conditions, such as the use of different fetal bovine serum, can also influence the morphology of astrocytes. In addition, the mechanical injury induced by the isolation procedures for astrocytes might contribute to variations in their morphology during in vitro cultivation. In summary, the morphological differences observed in GFAP+ astrocytes in cell culture likely result from a combination of culture conditions, cell density, and mechanical injury occured during astrocyte isolation etc.

- Additional verification of reactive astrocytes could be performed by different reactive astrocyte markers, such as GLAST, Sox9, S100ß. Thus, quantitative analysis of activated astrocytes can be done by counting DAPI vs GLAST, Sox9 or S100ß positive cells.

We really agree with the reviewer that there are other markers of reactive astrocytes such as GLAST, sox9 and S100β. However, numerous evidences support that GFAP is the most commonly used reactive astrocyte markers. Most of the cases, reactive astrocytes undergo GFAP overexpression. GFAP is one the most consistently induced gene in transcriptomic datasets of reactive astrocytes, confirming its usefulness as a reactive marker (Escartin et al., 2019). Thus, we have used GFAP as the marker of astrocyte activation in our study.

- How you performed quantifications for Figures 1C and 1D should be clearly explained, details are not given.

Quantification for Figure 1C and 1D were added in the figure legend. In general, Mean fluorescence intensity of PKM2 in different groups of (B) was calculated by ImageJ. The number of nuclear PKM2 was quantified by Image-Pro Plus software manually (eg. nuclear or cytoplasmic based on DAPI blue staining). The proportion of nuclear P KM2 is determined by normalizing the count of nuclear PKM2 to the count of nuclear DAPI, which represents the number of cell nuclei.

- "Together, these data demonstrated the nuclear translocation of PKM2 in astrocytes from EAE mice." Here the usage of "suggests" instead of "demonstrated".

Based on the reviewer's suggestion, we have revised the use of "demonstrated" to "suggest" in this sentence.

Result 2 and 3:- In the literature, DASA-58 is shown to be the activator of PKM2 (https://www.nature.com/articles/nchembio.1060, https://doi.org/10.1016/j.cmet.2019.10.015).- Providing references for the inhibitory use of DASA-58 for PKM2 would be appreciated.

DASA-58 is referred to as “PKM2 activator” due to its ability to enforce the tetramerization of PKM2, enhancing the enzymatic ability of PKM2 to catalyze PEP to pyruvate conversion. However, the enforced conversion of tetramerization of PKM2 inhibited the dimer form of PKM2, thereby inhibiting its nuclear translocation. For this reason, DASA-58 is also used as the inhibitor of nuclear translocation of PKM2. In primary BMDMs, LPS induced nuclear PKM2. However, driving PKM2 into tetramers using DASA-58 and TEPP-46 inhibited LPS-induced PKM2 nuclear translocation (Palsson-McDermott et al., 2015). Consistently, FSTL1 induced PKM2 nuclear translocation was inhibited by DASA-58 in BMDMs (Rao et al., 2022). Accordingly, we have added these references in the manuscript.

- Western blot results and statistics for PKM2 should be quantitatively given for all groups.

According to the reviewer’s suggestions, we have added the quantification of PKM2 for western blots in figure 2 and figure 3. Quantification of PKM2 in figure 2D is added in Fig S3. Quantification of PKM2 in figure 3D is added in Fig.S4B and Fig. S4C.

- Figure 3A-B: staining method/details are not mentioned in materials and methods.

Staining methods is in the paragraph entitled “Immunofluorescence” in the section of materials and methods. The descriptions are as follows:

For cell immunochemistry, cells cultured on glass coverslips were fixed with 4% PFA for 10 min at RT, followed by permeabilization with 0.3% Triton X-100. Non-specific binding was blocked with buffer containing 3% BSA for 30 min at RT. Briefly, samples were then incubated with primary antibodies and secondary antibodies. DAPI was used to stain the nuclei. Tissues and cells were observed and images were acquired using an EVOS FL Auto 2 Cell image system (Invitrogen). The fluorescence intensity was measured by ImageJ.

- In Figure 3A, in only DASA-58 treated cells, it looks like GFAP staining is decreased. It would be better to include MFI analysis for GFAP in the supplementary information.

We have added the MFI analysis for GFAP in Figure 3A in Fig.S4A. GFAP expression is decreased after DASA-58 treatment (in both control and MOGsup condition), the reason might be due to the effect of DASA-58 on inhibition of PKM2 nuclear transport, which subsequently suppress the activation of astrocytes, leading to the decreased expression of GFAP.

Result 4- Detailed explanation of the mass spectrometry and IP experiments should be given in materials and methods. What are the conditions of the cells? Which groups were analyzed? Are they only MOG stimulated, MOG-DASA-58 treated, or only primary astrocytes without any treatment? The results should be interpreted according to the experimental group that has been analyzed.

We have added the detailed information relating to mass spectrometry and immunoprecipitation in the materials and methods. In general, two groups of cells were subjected to mass spectrometry analysis, primary astrocytes without any treatment and MOGsup-stimulated primary astrocytes. These two groups were immunoprecipitated with anti-PKM2 antibody. Moreover, in the manuscript, we have revised the sentence concerning the description of mass spectrometry. The description is as follows: “To illustrate underlying mechanism accounting for nuclear translocation of PKM2 in astrocytes, we sought to identify PKM2-interacting proteins. Here, unstimulated and MOGsup-stimulated primary astrocytes were subjected to PKM2 immunoprecipitation, followed by mass spectrometry”. Furthermore, the description of these two groups of cells were added in the figure legend of Fig.4.

Result 5:- For the reader, it would be better to start this part by explaining the role of TRIM21 in cells by referring to the literature.

We agreed with the reviewer that beginning this part by explaining the role of TRIM21 would be better. Accordingly, we have added the following descriptions at the beginning of this part: “TRIM21 is a multifunctional E3 ubiquitin ligase that plays a crucial role in orchestrating diverse biological processes, including cell proliferation, antiviral responses, cell metabolism and inflammatory processes (Chen X. et al., 2022).” The relevant literature has been included: Chen X, Cao M, Wang P, Chu S, Li M, Hou P, et al. The emerging roles of TRIM21 in coordinating cancer metabolism, immunity and cancer treatment. Front Immunol 2022;13:968755.

- The source and the state of the cells (control vs MOG induced) should be stated (Figure 5A).

In figure 5A to 5D, single-cell RNA-seq were performed from CNS tissues of naive and different phases of EAE mice (peak and chronic). We have added this detailed information in the figure legend of Figure 5.

- Figure 5D can be placed after 5A. Data in Figure 5A is probably from naive animals, if so, it should be stated in the legend where A is explained. The group details of the data shown in Figure 5 should be clearly stated.

According to the reviewer’s suggestions, we have placed 5D after 5A. Single-cell RNA seq analysis were performed from CNS tissues of naïve mice and EAE mice. This information is stated in the legend of Figure 5A-D. “Single-cell RNA-seq profiles from naive and EAE mice (peak and chronic phase) CNS tissues. Naive (n=2); peak (dpi 14–24, n=3); chronic (dpi 21–26, n=2).”

- Immunofluorescence images should be replaced with better quality images, in control images, stainings are not visible.

We have replaced with better quality images in figure 5H and in control images, the staining is now visible.

Result 6:- Experimental procedures should be given in detail in materials and methods.

We have revised the section of materials and methods, and more details are added. Detailed information was added for astrocyte isolation, immunoprecipitation. Moreover, mass spectrometry, Hematoxylin-Eosin (HE) and Luxol Fast Blue (LFB) staining, Splenocyte isolation and supernatant of MOG35-55-stimulated-splenocytes were added in materials and methods.

Result 7:- In Figure 7A, the mean clinical score seems significantly reduced in the shTRIM21-treated group, although it is explained in the result text that it is not significant. Explain to us the difference between Figure 7A and the explaining text?

Thank you for pointing this out. We sincerely apologize for our carelessness. Based on your comments, we have made the corrections in the manuscript. As there is indeed a statistical difference in the mean clinical scores between shTRIM21-treated group and shVec group, we have accordingly revised the sentence for Figure 7A to state, “At the end time point at day 22 p.i., shTRIM21-treated group showed reduced disease scores compared to control groups (Fig. 7A).” .

- The staining methods for luxury fast blue and HE are not given in materials and methods.

According to the reviewer’s comments, we have added the staining methods for HE and LFB in materials and methods.

- In Figure 7E, authors claim that MBP staining is low in an image, however the image covers approximately 500 um area. One would like to see the demyelinated areas in dashed lines, and also the whole area of the spinal cord sections.

In Author response image 2, we have added the images for MBP staining of the whole area of spinal cord sections. Demyelinated areas are marked with dashed lines.

- "TEPP-46 is an allosteric activator that blocks the nuclear translocation of PKM2 by promoting its tetramerization." should be supported by references.

We have added two references for this sentence. Anastasiou D et al. showed that TEPP-46 acts as an activator by stabilizing subunit interactions and promoting tetramer formation of PKM2. Angiari S et al. showed that TEPP-46 prevented the nuclear transport of PKM2 by promoting its tetramerization in T cells.

These two references are added:

Angiari S, Runtsch MC, Sutton CE, Palsson-McDermott EM, Kelly B, Rana N, et al. Pharmacological Activation of Pyruvate Kinase M2 Inhibits CD4(+) T Cell Pathogenicity and Suppresses Autoimmunity. Cell metabolism 2020;31(2):391-405.e8.

Anastasiou D, Yu Y, Israelsen WJ, Jiang JK, Boxer MB, Hong BS, et al. Pyruvate kinase M2 activators promote tetramer formation and suppress tumorigenesis. Nature chemical biology 2012;8(10):839-47.

- Could you explain what the prevention stage is?

The term “prevention stage” was used to describe the administration of TEPP-46 before disease onset. To be more accurate, we have revised the phrase from “prevention stage” to “preventive treatment” as described in other references. For example, Ferrara et al. (Ferrara et al., 2020) used “preventive” and “preventive treatment” to mean administration before disease onset.

The revised sentences are as follows: “To test the effect of TEPP-46 on the development of EAE, the “preventive treatment” (i.e, administration before disease onset) was administered. Intraperitoneal treatment with TEPP-46 at a dosage of 50 mg/kg every other day from day 0 to day 8 post-immunization with MOG35-55 resulted in decreased disease severity (Fig. S8A).”

- In in vitro experiments, authors used DASA-58, and in vivo they used TEPP-46. What might be the reason that DASA-58 is not applied in vivo?

The effects of DASA-58 and TEPP-46 in promoting PKM2 tetramerization have been tested in vitro and has been documented. Based on in vitro absorption, distribution, metabolism and excretion profiling studies, Anastasiou et al. predicted that TEPP-46 had better in vivo drug exposure compared to DASA-58. Moreover, TEPP-46, but not DASA-58, is pharmacokinetically validated in vivo (Anastasiou et al., 2012). Thus, we used TEPP-46 for in vivo studies.

- Authors claim that TEPP-46 activates PKM2 and leads it its nuclear translocation, however, they did not verify PKM2 expression in the nucleus.

To support that TEPP-46 exerts effects in inhibiting PKM2 nuclear translocation both in vivo and in vitro, we have performed western blotting analysis and immunofluorescence staining. In vitro, TEPP-46 administration inhibited the MOGsup-induced PKM2 nuclear translocation, which exerts similar effects as DASA-58 (Author response image 4). The in vivo effects of TEPP-46 was analyzed by co-immunostaining of PKM2 and GFAP. The results showed reduced nuclear staining of PKM2 in spinal cord astrocytes in TEPP-46-treated EAE mice compared with control EAE mice (Figure S7B).

**Author response image 4. sa5fig4:** TEPP-46 inhibited the nuclear transport of PKM2 in primary astrocytes. Nuclear-cytoplasmic protein extraction analysis showed the nuclear and cytoplasmic changes of PKM2 in TEPP-46 treated astrocytes and MOGsup-stimulated astrocytes. Primary astrocytes were pretreated with 50 μM TEPP-46 for 30 min and stimulated with MOGsup for 24 h.

Supplementary Figure 3:- In Figure 3D, merge should be stated on top of the merged images, it is confusing to the reader.

According to the reviewer’s comments, we have added merge on top of the merged images.

Discussion:All results should be discussed in detail by interpreting them according to the literature.

We have further discussed the results in the discussion n section. Firstly, we added a paragraph describing the role of nuclear translocation of PKM2 in diverse CNS diseases. Moreover, a paragraph discussing the nuclear function of PKM2 as a protein kinase or transcriptional co-activator was added. Now the discussion section is more comprehensive, which nearly discuss all the results by interpreting them according to the literature in detail.

**Reviewer #2 (Recommendations For The Authors):**
The authors could address the following points:(1) In Figure 1A, the authors present immunofluorescence staining of PKM2 in both control mice and MOG35-725 55-induced EAE mice across different stages of disease progression: onset, peak, and chronic stages. Observing the representative images suggests a notable increase in PKM2 levels, particularly within the nucleus of MOG35-725 55-induced EAE mice. However, to provide a more comprehensive analysis, it would be beneficial for the authors to include statistical data, such as average intensities {plus minus} standard deviation (SD), along with the nuclear PKM2 ratio, akin to the presentation for cultured primary astrocytes in vitro in panels B-D. Additionally, the authors should clearly specify the number of technical repeats and the total number of animals utilized for these data sets to ensure transparency and reproducibility of the findings.

Thanks for the reviewer’s suggestion. Accordingly, for figure 1A, we have added the nuclear PKM2 ratio in astrocytes in control and different stages of EAE mice in Supplementary figure S1A. Moreover, the quantification of mean fluorescence intensity (MFI) for PKM2 was added in figure S1B. Moreover, we have added the number of animals used in each group in figure legend.

(2) The blue hue observed in the merged images of Figure 1B (lower panel) presents a challenge for interpretation. The source of this coloration remains unclear from the provided information. Did the authors also include a co-stain for the nucleus in their imaging? To enhance clarity, especially for individuals with color vision deficiency, the authors might consider utilizing different color combinations, such as presenting PKM2 in green and GFAP in magenta, which would aid in distinguishing the two components. Furthermore, for in vitro cell analysis, incorporating a nuclear stain could provide valuable insights into estimating the cytosolic-to-nuclear ratio of PKM2.

For the question relating to the merged images in figure 1B, PKM2 was presented in green, GFAP was presented in red and blue represents the nuclear staining by DAPI. “Merge” represents the merged images of these three colors. To enhance the clarity, we have added the images for the nuclear staining of DAPI.

(3) To substantiate the conclusion of the authors regarding the enhancement of aerobic glycolysis due to PKM2 expression and nuclear translocation in MOGsup-stimulated astrocytes, employing supplementary methodologies such as high-resolution respirometry and metabolomics could offer valuable insights. These techniques would provide a more comprehensive understanding of metabolic alterations and further validate the observed changes in glycolytic activity.

While we recognize the merits of techniques such as high-resolution respirometry and metabolomics, we believe that the conclusions regarding the enhancement of aerobic glycolysis due to PKM2 expression and nuclear translocation in MOGsup-stimulated astrocytes are sufficiently supported by the current experimental evidence. Our study has relied on a robust set of experiments, including lactate production, glucose consumption, cyto-nuclear localization analysis and western blotting analysis of key enzymes in glycolysis. These results, in conjunction with the literature on the role of PKM2 in various cancer cells, keratinocytes and immune cells, provide a strong foundation for our conclusions. Although metabolomics could offer a global view of the changes in metabolic states in astrocytes, as the end product of aerobic glycolysis is lactate, our study, which analyze the change of lactate levels in different experimental conditions might be more direct. However, we fully acknowledge that future studies employing these advanced methodologies could provide further insights into the precise mechanisms underlying PKM2's effects on aerobic glycolysis.

(4) Minor: Why is the style of the columns different in Gig 2 panel D compared to those shown in panels B, C, and G of Figure 2.

To maintain consistency in the column style across figure 2, we have updated the column in figure 2D. Now, we use same style of columns in Fig 2B, C, D and G.

(5) The effect of stimulating astrocytes with MOGsup on cell proliferation, as shown in Figure 2E, is very moderate. Does DASA-58 reduce the proliferation of control cells in this assay?

In response to the reviewer’s questions, we conducted a CCK8 analysis in astrocytes subjected to DASA-58 treatment. As depicted in Author response image 5, administration of DASA-58 did not reduce the proliferation of control cells. This result aligns with our other findings in the glycolysis assays and EdU analysis, where there is no statistical difference between control group and DASA-58-treated group. One plausible explanation for this is that in their steady state, astrocytes in the control group are not in a hyperproliferative state. Under such conditions, inhibiting the translocation of PKM2 via DASA-58 or other inhibitors did not significantly affect the proliferation of astrocytes.

**Author response image 5. sa5fig5:** CCK8 analysis of astrocyte proliferation. Primary astrocytes were pretreated with 50 μM DASA-58 for 30 min before stimulation with MOGsup. Data are represented as mean ± SEM. ***P<0.001. SEM, standard error of the mean.

(6) The tables and lists in Figure 4, panels A-D, are notably small, hindering readability and comprehension. Consider relocating these components to the supplementary materials as larger versions.

We have updated the tables and lists, the lines are made thicker. As suggested by the reviewer, we relocate theses components in Supplementary Figure S5.

**Reviewer #3 (Recommendations For The Authors):**
Higher magnification images that more clearly show nuclear translocation of PKM2 and pp65 and pSTAT3 immunoreactivity should be added to the figures panels, for example as inlets.

Thank you for pointing out this issue in the manuscript. According to the reviewer’s comments we have included higher magnification images as inlets for Figure 3A, Figure 3B and Figure 2A. These enlarged images now provide a clearer visualization of the nuclear translocation state of PKM2, pp65, and pSTAT3.

There are seldom wording errors like features => feathers at line 364.

We are very sorry for our incorrect writing. We have corrected this spelling mistake in the manuscript.

**Reviewer #4 (Recommendations For The Authors):**
Here below are major and minor concerns on the data presented:(1) It is not clear from the Methods section what are the culture conditions defined as 'control' in Figure 1B-D. I believe the control should be culturing with the conditioned medium of normal (non-EAE) mice splenocytes to be sure the effect is not from cytokines naturally secreted by these cells.

Thanks for the reviewer’s comments and we totally understand the reviewer's concern. The control means non-treated primary astrocytes cultured with traditional DMEM medium supplemented with 10% FBS. In fact, we have performed experiments to exclude the possibility that the observed effect of MOGsup on the activation of astrocytes is from cytokines secreted by splenocytes. Splenocytes from normal (non-EAE) mice were isolated, cultured in RPMI-1640 medium containing 10% FBS for 60 hours, and supernatant was collected. Immunofluorescence staining of PKM2 and GFAP were performed in non-treated primary astrocytes and astrocytes stimulated with supernatant from control splenocytes. As shown in Figure S1C, in both groups, no difference was observed in PKM2 expression and localization, PKM2 was located mainly in the cytoplasm in theses conditions. These results indicate that observed effect of PKM2 in MOGsup-stimulated condition is not due to the cytokines secreted from splenocytes. Thus, we used non-treated primary astrocytes as controls in our study. To clarify the control group, we have revised the description in the figure legend, The revised expression is as follows: “Immunofluorescence staining of PKM2 (green) with GFAP (red) in non-treated primary astrocytes (control) or primary astrocytes cultured with splenocytes supernatants of MOG35–55-induced EAE mice (MOGsup) for different time points (6 h, 12 h and 24 h). ”

(2) Figure 3D: the presence of PMK2 in the nuclear fraction upon MOGSUP together with the DASA-58 (last lane of Figure 3D) is not supporting the hypothesis proposed and further may indicate that the reduction of pSTAT3, pp65, etc. observed is independent of PMK2 nuclear translocation/astrocyte activation being observed even in absence of MOGSUP.

Thank you for pointing out this problem in manuscript. The representing image of nuclear level of PKM2 in Figure 3D is not obvious, as shown by figure 3D, which has raised doubts among the reviewers. To strengthen our conclusion that the reduction of STAT3 and p65 pathway is related to the inhibited nuclear level of PKM2 induced by DASA-58, nuclear PKM2 level was quantified and added in Figure S4B. From the quantification results, it is evident that DASA-58 administration decreased the nuclear level of PKM2 in MOGsup-stimulated astrocytes. To address this concern, we have updated the immunoblot image for PKM2 in figure 3D and incorporated quantification results in supplementary Figure S4.

(3) Molecular docking indication and deletion co-immunoprecipitation reported in Figure 4 data are not concordant on TRIM21: N-terminal Phe23 and Thr87 (Figure 4E) predicted by MD to bind PMK2 are not in the PRY-SPRY domain suggested by the co-IP experiment (Figure 4I).

The discrepancy between the molecular docking prediction and the co-immunoprecipitation can be explained as follows:

Firstly, molecular docking is computational methods that predicts protein-protein interaction based on 3-D structures of the proteins. However, the accuracy of this predication can be influenced by the different models of 3D structures of TRIM21 and PKM2, as well as by factors such as post-translational modifications and flexibility of the proteins. Proteins in vivo are subject to post-translational modifications that can affect their interactions. These modifications are not fully captured in molecular docking analysis. For example, in our analysis, the predicted N-terminal Phe23 and Thr87 in TRIM21 hold the potential to interact with PKM2 by hydrogen bonds. However, such binding can be influenced by diverse biological environments, such as different cells and pathological conditions. Molecular docking predication may suggest the specific residues and binding pocked within the protein complex, however, the accuracy should be verified by experimental techniques such as immunoprecipitation. To address the predication results of molecular docking, the description has been revised as follows: “TRIM21 is predicted to bound to PKM2 via hydrogen bonds between the amino acids of the two molecules.”

Co-immunoprecipitation that involves the use of truncated domains of TRIM21 and PKM2, is an experimental technique relies on the specific interaction between antibody and targeted proteins. This technique can provide insights into the precise binding domains between TRIM21 and PKM2. As demonstrated in our study, PRY-SPRY domain of TRIM21 is involved in this binding. In summary, while molecular docking and Co-IP are valuable tools for studying protein-protein interactions, their differing focus and limitations may result in discrepancies between the predicted interaction sites and the experimentally identified interaction domains.

(4) The Authors state that PMK2 is a substrate of TRIM21 E3 ligase activity, however, this is not proved: (i) interaction does not imply a ligase-substrate relationship; (ii) the ubiquitination shown in Figure 6C is not performed in denaturing conditions thus the K63-Ub antibody can detect also interacting FLAG-IPed proteins (besides, only a single strong band is seen, not a chain; molecular weights in immunoblot should be indicated); (iii) use of a catalytically inactive TRIM21 would be required as well.

We appreciate the reviewer’s comments regarding the limitations of the immunoprecipitation and K63-antibody test, which could not lead to the conclusion that PKM2 is a substrate of TRIM21. To avoid any misunderstandings, we have revised the relevant sentence from “Hereby, we recognized PKM2 as a substrate of TRIM21” to “Hereby, we recognized PKM2 as an interacting protein of TRIM21, and further studies are required to determine if it is a substrate of E3 ligase TRIM21”. Moreover, we have revised the title of the relevant part in the results section, the previous title, “TRIM21 ubiquitylates and promotes the nuclear translocation of PKM2” has been replaced with “TRIM21 promotes ubiquitylation and the nuclear translocation of PKM2”. Moreover, molecular weights for all proteins in western blotting were indicated.

(5) As above, molecular weights should always be indicated in immunoblot.

Thanks for pointing out this problem in the figures. Accordingly, we have added the molecular weights for every protein tested in immunoblot.

(6) The authors should describe the EAE mouse model in the text and in the material and methods as it may not be so well known to the entire reader audience, and the basic principle of MOG35-55 stimulation, in order to understand the experimental plan meaning.

We appreciate the reviewer’s comments highlighting the importance of clarifying EAE model for a broader understanding of the reader audience. In response, we have described the EAE model both in the text and in the materials and methods section. In the text, the description of EAE model was added at the beginning of the first paragraph in the Results section. The description is as follows: “EAE is widely used as a mouse model of multiple sclerosis, which is typically induced by active immunization with different myelin-derived antigens along with adjuvants such as pertussis toxin (PTX). One widely used antigen is the myelin oligodendrocyte glycoprotein (MOG) 35-55 peptide (Nitsch et al., 2021), which was adopted in our current studies.”

We have also added the detailed experimental procedures for EAE induction in the materials and methods section.

(7) The authors should better explain and give the rationale for the use of splenocytes and why directly activated astrocytes (isolated from the EAE model) cannot be employed to confirm/prove some of the presented data.

Firstly, splenocytes offer a heterogenous cell population, encompassing T cells and antigen presenting cells (APC), which may better mimic the microenvironment and complex immune responses observed in vivo.

Myelin oligodendrocyte glycoprotein (MOG) 35-55 peptide is one widely used antigen for EAE induction. MOG35-55 elicits strong T responses and is highly encephalitogenic. Moreover, MOG35-55 induces T cell-mediated phenotype of multiple sclerosis in animal models. Thus, by isolating splenocytes from the onset stage of EAE mice, which contains APC and effector T cells, followed by stimulation with antigen MOG35-55 in vitro for 60 hours, the T-cell response in the acute stage of EAE diseases could be mimicked in vitro. The supernatant from MOG35-55 stimulated splenocytes has high levels of IFN-γ and IL-17A, which in part mimic the pathological process and environment in EAE, and this technique has been documented in the references (Chen et al., 2009, Kozela et al., 2015).

Correspondingly, we have revised sentence for the use of MOG35-55 stimulates splenocytes in EAE mice and add the relevant references: “Supernatant of MOG35-55-stimulated splenocytes isolated from EAE mice were previously shown to elicit a T-cell response in the acute stage of EAE and are frequently used as an in vitro autoimmune model to investigate MS and EAE pathophysiology (Chen et al., 2009, Du et al., 2019, Kozela et al., 2015).”

Secondly, activated astrocytes (isolated from the EAE model) can not be employed for in vitro culture for the following reasons:

(1) Low cell viability. Compared to embryonic or neonatal mice, adult mice yield a limited number of viable cells. The is mainly because that adult tissues possess less proliferative capacity.

(2) Disease changes. Astrocytes in EAE mice are exposed to microenvironment including inflammatory cytokines, antigens and other pathological factors. Without this environment, the function and morphology of astrocytes undergo changes, which make it difficult to interpret the results in vitro.

For these reasons, the in vitro cultured primary astrocytes used the neonatal mice.

(8) The authors should indicate the phosphorylation sites they are referring to when analysing p-c-myc, pSTAT3, pp65, etc...

According to the reviewer’s suggestions, we have added the phosphorylation sites for pSTAT3 (Y705), pp65 (S536), p-c-myc (S62) and pIKK (S176+S180) in the figure panels.

(9) Reference of DASA-58 and TEPP-46 inhibitors and their specificity should be given.

According to the reviewer’s comments, we have added the relevant references for the use of DASA-58 and TEPP-46 as inhibitors of PKM2 nuclear transport. In primary BMDMs, LPS induced nuclear PKM2. However, driving PKM2 into tetramers using DASA-58 and TEPP-46 inhibited LPS-induced PKM2 nuclear translocation (Palsson-McDermott et al., 2015). Consistently, FSTL1 induced PKM2 nuclear translocation was inhibited by DASA-58 in BMDMs (Rao et al., 2022). Accordingly, we have added these references in the manuscript.

To address the selectivity of TEPP-46 and add the references, the relevant sentence has been revised from “TEPP-46 is an allosteric activator that blocks the nuclear translocation of PKM2 by promoting its tetramerization” to “TEPP-46 is a selective allosteric activator for PKM2, showing little or no effect on other pyruvate isoforms. It promotes the tetramerization of PKM2, thereby diminishing its nuclear translocation (Anastasiou et al., 2012, Angiari et al., 2020).”

Reviewing Editor (Recommendations For The Authors):The reviewing editor would appreciate it if the original blots from the western blot analysis, which were used to generate the final figures, could be provided.

Thanks for the reviewing editor’s comment, accordingly, we will add the original blots for the western blots analysis.

References

Anastasiou D, Yu Y, Israelsen WJ, Jiang JK, Boxer MB, Hong BS, et al. Pyruvate kinase M2 activators promote tetramer formation and suppress tumorigenesis. Nature chemical biology 2012;8(10):839-47.

Escartin C, Guillemaud O, Carrillo-de Sauvage M-A. Questions and (some) answers on reactive astrocytes. Glia 2019;67(12):2221-47.

Ferrara G, Benzi A, Sturla L, Marubbi D, Frumento D, Spinelli S, et al. Sirt6 inhibition delays the onset of experimental autoimmune encephalomyelitis by reducing dendritic cell migration. Journal of neuroinflammation 2020;17(1):228.

Lin CC, Edelson BT. New Insights into the Role of IL-1β in Experimental Autoimmune Encephalomyelitis and Multiple Sclerosis. Journal of immunology (Baltimore, Md : 1950) 2017;198(12):4553-60.

Palsson-McDermott Eva M, Curtis Anne M, Goel G, Lauterbach Mario AR, Sheedy Frederick J, Gleeson Laura E, et al. Pyruvate Kinase M2 Regulates Hif-1α Activity and IL-1β Induction and Is a Critical Determinant of the Warburg Effect in LPS-Activated Macrophages. Cell metabolism 2015;21(1):65-80.Rao J, Wang H, Ni M, Wang Z, Wang Z, Wei S, et al. FSTL1 promotes liver fibrosis by reprogramming macrophage function through modulating the intracellular function of PKM2. Gut 2022;71(12):2539-50.

Wheeler MA, Clark IC, Tjon EC, Li Z, Zandee SEJ, Couturier CP, et al. MAFG-driven astrocytes promote CNS inflammation. Nature 2020;578(7796):593-9.

Zhang J, Feng G, Bao G, Xu G, Sun Y, Li W, et al. Nuclear translocation of PKM2 modulates astrocyte proliferation via p27 and -catenin pathway after spinal cord injury. Cell Cycle 2015;14(16):2609-18.